# The sympathoregulatory region of the mouse rostral brainstem relies on both GABA and glycine to generate inhibitory currents

Hong Gao[1], Lucie D. Desmoulins[1], Adrien J. R. Molinas[1], Courtney M. Dugas[1], Andrea Zsombok[1,2] and Andrei V. Derbenev[1,2]

[1]*Department of Physiology, School of Medicine, Tulane University, New Orleans, LA, USA*
[2]*Brain Institute, Tulane University, New Orleans, LA, USA*

Handling Editors: Katalin Toth & Samuel Young

The peer review history is available in the Supporting information section of this article (https://doi.org/10.1113/JP288116#support-information-section).

**Abstract figure legend** Our study demonstrates that both GABA and glycine mediate inhibitory postsynaptic currents in the ventrolateral/ventromedial medulla (VLM/VMM) and increasing the activity of inhibitory synapses facilitates glycine release. We found that presympathetic VLM/VMM neurons receive glycinergic inputs, which release both glycine and GABA, and inhibition of glycine transporter 2 (GlyT$_2$) decreased glycine release. In summary, our study suggests that in the mouse brainstem, glycinergic neurons release both GABA and glycine, and these inhibitory mechanisms may be equally important for the regulation of presympathetic neurons.

**Abstract** Presympathetic neurons in the ventrolateral and ventromedial medulla (VLM/VMM) control sympathetic tone, and GABA and glycine were identified as inhibitory neurotransmitters. Although glycinergic inhibition of presympathetic neurons in the VLM/VMM has been shown, the sources of glycinergic inputs and the mechanism of glycine release are ill-defined. Here, we tested whether presympathetic neurons in the VLM/VMM receive glycinergic inputs. Glycine

**Hong Gao** received her PhD from the Department of Cell and Molecular Biology at Tulane University in 2008 where she studied the inhibitory regulation of neurons of the rat dorsal motor nucleus of vagus. Currently, she works as a research scientist in the Department of Physiology at the Tulane University School of Medicine. Her studies focus on organization and synaptic regulation of presympathetic neurons in the rostral brainstem.

is recycled by glycine transporter 2 (GlyT$_2$), which is a reliable marker of glycinergic neurons. GlyT$_2^{\text{Cre}}$ mice were cross bred with channelrhodopsin (ChR2) mice to generate GlyT$_2^{\text{ChR2/EYFP}}$ mice to be able to selectively stimulate GlyT2-expressing fibres. Presympathetic neurons in the VLM/VMM were retrogradely labelled with pseudorabies virus and whole-cell, patch clamp recordings were conducted in brainstem slices. Our study demonstrates that inhibitory synaptic currents recorded from presympathetic VLM/VMM neurons are mediated by GABA and glycine. We found that increasing the activity of inhibitory synapses facilitates glycine release. Light stimulation of GlyT$_2^{\text{ChR2/EYFP}}$ fibres triggered monosynaptic evoked currents composed of GABA and glycine and we found that sustained glycine release depends on glycine uptake mediated by GlyT$_2$. In addition, we used a combination of monosynaptic and transsynaptic viral tracings to identify the location of presympathetic glycinergic neurons in the ventral brainstem, and immunostaining to reveal the expression of GlyR $\alpha$1, $\alpha$3 and $\beta$ subunits in VLM/VMM neurons. Our study identified that GlyT2-expressing neurons rely on release of both GABA and glycine to generate inhibitory currents. However, the potential role and the necessity for both neurotransmitters in the control of presympathetic VLM/VMM neurons require additional studies.

(Received 15 November 2024; accepted after revision 26 January 2026; first published online 20 February 2026)

**Corresponding author** A. V. Derbenev: Department of Physiology, Tulane University, School of Medicine, New Orleans, LA, USA.    Email: aderben@tulane.edu

### Key points

- Presympathetic neurons in the mouse ventrolateral/ventromedial medulla (VLM/VMM) receive monosynaptic glycinergic inputs that release both GABA and glycine.
- Glycinergic neurons were identified in the lateral paragigantocellular nucleus and Bötzinger complex.
- Increasing the activity of inhibitory synapses facilitates glycine release and sustained glycine release depends on glycine uptake mediated by glycine transporter 2 (GlyT2).
- Three prevalent forms of glycine receptors ($\alpha$1, $\alpha$3 and $\beta$) are expressed in the VLM/VMM.
- Our study demonstrates the necessity of GABA and glycine for the regulation of presympathetic neurons and thus highlights the need for a better understanding of glycinergic circuits.

## Introduction

GABA and glycine are the major inhibitory neurotransmitters in the CNS. Both neurotransmitters are remarkably versatile as they act on ionotropic receptors (Cl$^-$ channels) leading to hyperpolarization and inhibition of neuronal activity (Burger et al., 1991; Wojcik et al., 2006). Postsynaptic GABAergic and glycinergic receptors (GABA$_A$R and GlyR) are often clustered together (Fischer et al., 2000; Kneussel & Betz, 2000; Levi et al., 1999), and accumulating evidence shows that GABA and glycine are co-released in the spinal cord, dorsal cochlear nucleus, abducens nucleus, ventrolateral/ventromedial medulla (VML/VMM) and the medial nucleus of the trapezoid body (Aubrey & Supplisson, 2018; Dun & Mo, 1989; Gao et al., 2019; Lu et al., 2008; Moore & Trussell, 2017; Russier et al., 2002; Werynska et al., 2023). Various roles for reliance on both GABA and glycine have been proposed including

the fine inhibitory tuning of the neurons via separate receptor activation (Kuo et al., 2009; Moore & Trussell, 2017; Russier et al., 2002; Xie & Manis, 2013), extra-fast inhibition (Lu et al., 2008), compensation of one neurotransmitter by the other (Ishibashi et al., 2013; Nerlich et al., 2014), and biphasic inhibition where GABA controls neuronal threshold excitability and glycine controls the strength of inhibition (Gao et al., 2019).

Presympathetic neurons in the ventral brainstem (VLM/VMM) relay information to sympathetic preganglionic neurons in the spinal cord and thus play a crucial role in the regulation of sympathetic nerve activity (SNA). In the rostral VLM (RVLM), blockade of GABA$_A$R but not GlyR increases SNA (Amano & Kubo, 1993; Blessing, 1988; Cravo & Morrison, 1993; Guyenet et al., 1990; Heesch et al., 2006; Sun & Guyenet, 1985), suggesting that presympathetic neurons are under sustained GABAergic inhibition. Earlier studies conducted in rats showed that stimulation of the

gigantocellular nucleus with microinjection of glutamate causes hypotension. The drop in blood pressure was proposed to be the result of stimulation of inhibitory presynaptic neurons and GABAergic and glycinergic neurons were proposed to be located in a portion of the gigantocellular depressor area (Aicher et al., 1994). Intriguingly, it has been shown that local microinjection of glycine into the RVLM decreases SNA and mean arterial pressure (Sakima et al., 2000), suggesting that glycine acts as an inhibitory neurotransmitter and plays a role in the regulation of SNA. Anatomical studies have identified a large number of GABAergic and glycinergic neurons containing both $GlyT_2$ and GAD-67 mRNA in the VLM/VMM, and an electron microscopic investigation has shown that synaptic terminals on presympathetic neurons contained one or more amino acids, including glycine and GABA (Hossaini et al., 2012; Llewellyn-Smith et al., 2001; Schreihofer et al., 2000; Stornetta et al., 2004). Moreover, slice electrophysiology revealed that bath application of GABA or glycine increases membrane conductance, hyperpolarizes presympathetic RVLM neurons and decreases their firing rate (Gao & Derbenev, 2013; Gao et al., 2019).

In the current study, we used a combination of neuroanatomical and electrophysiological approaches to test whether presympathetic neurons in the VLM/VMM receive glycinergic inputs and if glycine release is increased when inhibitory activity increases. Our study demonstrates that presympathetic neurons in the VLM/VMM receive monosynaptic glycinergic inputs that release both inhibitory neurotransmitters, GABA and glycine. We found that glycine release is increased when the activity of inhibitory synapses increases and the sustainability of glycine release depends on glycine uptake mediated by $GlyT_2$. In addition, we used a combination of monosynaptic and transsynaptic viral tracings to identify presympathetic glycinergic neurons in the VLM/VMM, and immunostaining to reveal the expression of GlyR $\alpha 1$, $\alpha 3$ and $\beta$ subunits in these neurons. Our study demonstrates the existence of GABA and glycine release from $GlyT_2$-expessing neurons and indicates the necessity of both neurotransmitters for the regulation of presympathetic neurons, and thus underscores the need for a better understanding of glycinergic circuits.

## Methods

### Ethics

All experiments were approved by the Institutional Animal Care and Use Committee of Tulane University (protocol ID 2207). C. Steele, Interim Vice President for Research, Institutional Official & Research Integrity Officer, is responsible for Research Governance at the institution where the research was carried out. Male and female mice were housed in an institutionally approved on-site animal care facility using a 14:10 h ratio of light/dark, with food and water available *ad libitum*. Animal housing and experiments were performed following the guidelines of the National Institutes of Health Guide for the Care and Use of Laboratory Animals.

Heterozygous $GlyT_2^{Cre}$ mice [Tg(Slc6a5-cre) KF109Gsat/Mmucd, MMRRC_030730-UCD] were cross bred with channelrhodopsin 2 mice [Ai32(RCL-ChR2(H134R)/EYFP, JAX: 024109], resulting in ChR2 expression in $GlyT_2$-expressing neurons ($GlyT_2^{ChR2/EYFP}$). Experiments were performed using male and female $GlyT_2^{ChR2/EYFP}$, 8–25-week-old mice, whereas control experiments were conducted using $GlyT_2$ wild-type littermates.

### Viruses

Pseudorabies virus 614 (PRV-614), a retrogradely transported viral vector strain isogenic with PRV-Bartha that shows enhanced red fluorescent protein (RFP), was supplied by the NCRR CNNV Virus Center (Pittsburgh, PA, USA). PRV-614 was used to identify presympathetic neurons in the VLM/VMM, as described previously (Gao & Derbenev, 2013; Gao et al., 2019).

A retrograde AAV vector that expresses enhanced green fluorescent protein (EGFP) in a Cre-dependent manner [pAAV-hSyn-DIO-EGFP, a gift from Bryan Roth (Addgene viral prep #50457-AAVrg; RRID:Addgene_50457)] was injected into the VLM/VMM of GlyT2 mice to identify glycinergic neurons.

### Surgical procedures

To identify presympathetic neurons in the VLM/VMM, PRV-614 was injected into the left kidney, as previously described (Gao & Derbenev, 2013; Gao et al., 2019; Jiang et al., 2013). Briefly, animals were given a subcutaneous injection of Torbugesic 1 mg/kg plus meloxicam 5 mg/kg to minimize pain. Then, under isoflurane anaesthesia (2–3%), a small dorsolateral incision was made to expose the left kidney and two injections (2 µl each) of PRV-614 ($1 \times 10^8$ plaque-forming units/ml) were made into the cortex of the left kidney with a glass pipette (tip diameter approximately 50 µm). After surgery, the animals recovered in a biosafety level 2 facility for ∼96 or ∼120 h.

To reveal the distribution of $GlyT_2$-expressing neurons in the ventral brainstem, pAAV-hSyn-DIO-EGFP was stereotaxically injected into ventral brainstem. Briefly, the animals were treated with a subcutaneous injection of Torbugesic 1 mg/kg plus meloxicam 5 mg/kg to minimize pain. Then, under isoflurane anaesthesia

(2–3%), the viral construct was injected into the ventral brainstem of GlyT$_2$$^{Cre}$ mice using a pulled glass micropipette connected to a nano-injector (Nanoject III, Drummond, Sci. Co., Broomall, PA, USA). A single injection of 125 nl of pAAV-hSyn-DIO-EGFP (1.00 × 10$^{13}$ vg/ml) was performed at a rate of 1 nl/s. The virus was delivered unilaterally into the left VLM (AP: −6.7 mm; ML: ±1.1 mm; DV: −6.3 mm). The coordinates were determined according to the mouse brain atlas (Franklin & Paxinos, 2007). To prevent back-flow, the micropipette remained in place for an additional 5 min. The incision site was closed using surgical nylon sutures and the animals recovered in a biosafety level 1 facility for 3 weeks.

### Slice electrophysiology

Mice were decapitated under isoflurane anaesthesia ∼96 h after injection with PRV-614. Transverse brainstem slices (300 μm thick) containing PRV-labelled neurons were made by a vibrating microtome (Leica VT 1200S, Wetzlar, Germany) and stored at 35–37°C in oxygenated (95% O$_2$ + 5% CO$_2$) artificial cerebrospinal fluid (ACSF). ACSF contained (in mM): 124 NaCl, 3 KCl, 26 NaHCO$_3$, 1.4 NaH$_2$PO$_4$, 11 glucose, 2 CaCl$_2$ and 1.3 MgCl$_2$ (pH 7.2–7.4). PRV-labelled neurons in the VLM/VMM were visualized and targeted for recording based on their fluorescence. The recorded neurons were *post hoc* identified using the avidin–biotin AMCA fluorescence reaction.

Whole-cell patch-clamp recordings from PRV-labelled presympathetic neurons in the VLM/VMM were performed using a Multiclamp 700B amplifier. Data were filtered at 10 kHz, digitized at 20 kHz via a Digidata 1440A and acquired by pClamp 10.5 software (Molecular Devices, Sunnyvale, CA, USA). Recording electrodes were pulled from borosilicate glass (KG-33, King Precision Glass, Claremont, CA, USA) by a horizontal puller (P-1000, Sutter Instrument, Novato, CA, USA). Recording pipettes were directed to the labelled neurons using infrared differential interference contrast optics (Eclipse FN1, Nikon) with a 40× water-immersion objective. Recording electrodes (2–6 MΩ) were filled with a solution containing (in mM): 130 caesium gluconate, 1 NaCl, 5 EGTA, 10 Hepes, 1 MgCl$_2$, 1 CaCl$_2$, 2–4 ATP and 0.1% biocytin (pH 7.2–7.4) (adjusted with CsOH). Liquid junction potential was corrected using the pipette offset amplifier function before each recording. Data were excluded from analysis if series resistance changed by 25% throughout the experiment.

Spontaneous inhibitory postsynaptic currents (sIPSCs) or light-evoked IPSCs (eIPSCs) were examined at a holding potential of −10 mV. This experimental condition permitted the exclusion of excitatory glutamatergic trans-mission without using glutamate receptor antagonists that are known to decrease the presynaptic release of GABA in the brainstem (Boychuk & Smith, 2016; Xu & Smith, 2015). Light-evoked IPSCs were measured in response to ChR2 wide-field stimulation by coupling a xenon lamp (Lumen 200, Prior) to the microscope epifluorescence port and delivering brief pulses of UV light.

### Neuropharmacological design of experiments

To separate GABAergic IPSCs from glycinergic IPSCs, bicuculline methiodide (10 μM) was used, whereas strychnine (1 μM) was used to inhibit glycinergic IPSCs. It is important to note that strychnine was shown to block GABA$_A$R-mediated currents (Braestrup & Nielsen, 1980); therefore, to eliminate interaction of strychnine with GABA$_A$Rs, first we blocked GABA$_A$Rs with bicuculline, then applied strychnine to confirm involvement of GlyRs. To increase network excitability, we used 4-aminopyridine (4-AP; 1 mM). 4-AP is a non-selective, voltage-dependent potassium channel blocker that prolongs the depolarization of neurons and increases the release of neurotransmitters (Thompson, 1977). TTX, a voltage-activated sodium channel blocker, was used to determine whether the inputs were mono- or polysynaptic (Petreanu et al., 2009). N-[[1-(dimethylamino)cyclopentyl]methyl]-3,5-dimethoxy-4-(phenylmethoxy)benzamide hydrochloride (ORG25543, 10 μM), a potent and selective GlyT$_2$ inhibitor, was used to prevent the transport of glycine back to the neurons. Bicuculline, strychnine, 4-AP, TTX and ORG25543 were dissolved in ACSF. The chemicals were obtained from Sigma-Aldrich (St Louis, MO, USA) or Tocris Bioscience (Ellisville, MO, USA).

### Experimental design and statistical analysis

Patch-clamp recordings were analysed using pClamp 10.5 (Molecular Devices) and Mini Analysis 6 (Synaptosoft) software. The effects of 4-AP, bicuculline and strychnine on the frequency and amplitude of sIPSCs were analysed within a recording by the Kolmogorov–Smirnov two-sample test (a non-parametric, distribution-free goodness-of-fit test for probability distributions), with at least a few minutes of continuous activity being measured for each condition, as described previously (Gao & Derbenev, 2013; Gao et al., 2019; Jiang et al., 2013). The mean phasic current (integrated current of IPSCs) was calculated using: $I_{phasic} = f \times Q$, where $f$ is the synaptic frequency and $Q$ is the charge transfer measured as the area under the IPSC (Gao et al., 2017; Gao et al., 2019; Nusser & Mody, 2002). The frequency, amplitudes and $I_{phasic}$ of IPSCs across the neuronal population were analysed using a paired, two-tailed

Student's *t* test for Gaussian distributed values or a two-tailed Wilcoxon matched-pair signed-rank test for non-Gaussian distributed values. In addition, multiple comparisons within the population were performed with one-way ANOVA for Gaussian distributed values or Friedman test, followed by Dunn's multiple comparisons test (Prism, version 10.0 GraphPad Software Inc., La Jolla, CA, USA) for non-Gaussian distributed values. Two-way ANOVA was used to analyse the comparisons for the amplitude and ratio of $P_1$ to $P_n$ induced by the train of light stimuli in two conditions. Figure 1 comparisons were corrected using the Holm–Bonferroni method (Holm, 1979). Data are presented as the mean $\pm$ SD. $P < 0.05$ was considered statistically significant.

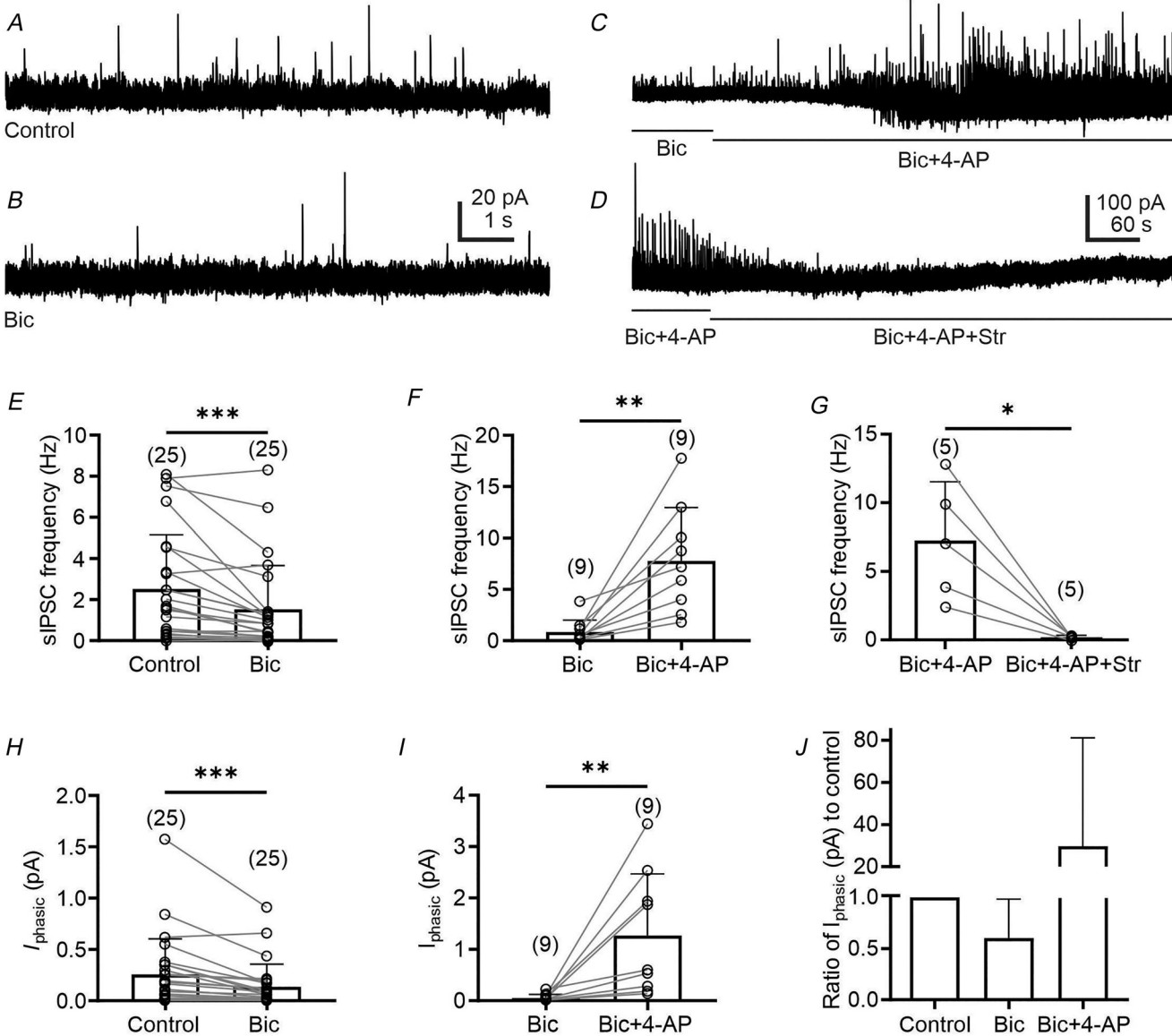

**Figure 1. Mixed GABAergic and glycinergic inhibitory currents in the VLM/VMM**
*A*, continuous recording of sIPSCs from a presympathetic neuron in the VLM/VMM voltage-clamped at −10 mV. *B*, same neuron as in *A* after application of the GABA receptor blocker, bicuculline (Bic, 10 µM). *C*, same neuron following co-application of 4-AP (1 mM) and Bic. In the presence of bicuculline, 4-AP increased sIPSC frequency. *D*, the same neuron in the presence of 4-AP and bicuculline before and after application of the GlyR blocker, strychnine (Str, 1 µM). These data confirmed that strychnine abolished the remaining sIPSCs. *E–J*, summary data showing the effect of bicuculline, 4-AP and strychnine on the frequency of sIPSCs (*E–G*) and $I_{phasic}$ (*H–J*) in presympathetic VLM/VMM neurons. The number of replicates is shown in parentheses. Bar graphs represent the mean $\pm$ SD; open circles represent individual data points. Statistical analyses were performed using the Wilcoxon matched-pairs signed-rank test (*E, F, H, I*) and paired *t* test (*G*). *P*-values were adjusted using the Holm–Bonferroni method. *$P < 0.05$, **$P < 0.01$, ***$P < 0.001$.

## Immunofluorescence staining and imaging

Mice were deeply anaesthetized with an injection of 80 mg/kg ketamine plus 5 mg/kg xylazine and transcardially perfused with phosphate-buffered saline (PBS) followed by 4% paraformaldehyde in 10 mM PBS (pH 7. 4). The brains were removed, postfixed for 24 h in the fixative solution, immersed in 30% sucrose until they equilibrated and sectioned at 35 μm on a cryostat (ThermoFisher Microm HM 550, Waltham, MA, USA).

Immunofluorescence staining of GlyR $\alpha$1, $\alpha$3 and $\beta$ subunits in presympathetic VLM/VMM neurons was performed using brainstem sections with 96 h PRV inoculation of the kidney. For the detection of GlyR$\alpha$1, heat-induced antigen retrieval was performed to improve the signal strength. After several rinses in PBS, floating sections were immersed in a 10 mM citrate solution (pH = 6) at 60°C overnight. After cooling, sections were rinsed several times in PBS before being transferred to blocking solution for 2 h (100 mM PBS with 0.3% Triton-X-100 and 5% normal donkey serum). For the GlyR$\alpha$3 detection, sections were mounted on slides and dried overnight. Using a Pap Pen (Vector Laboratories, Burlingame, CA, USA), a hydrophobic barrier was drawn around the sections to prevent reagent waste during the incubations. Signal strength was improved by enzymatic-induced antigen retrieval. Sections were incubated with a solution of 0.05% trypsin and 0.1% calcium chloride for 20 min at 37°C. After section cooling, several rinses in PBS were performed before sections were transferred into the blocking solution for 2 h. For the detection of GlyR $\beta$ subunit, floating sections were rinsed several times in PBS before being transferred into blocking solution.

Sections were then incubated with appropriate primary antibody prepared in the blocking solution: rabbit anti-GlyR alpha 1 (Synaptic System, #146-118, 1:200, Goettingen, Germany) or rabbit anti-GlyR alpha 3 (Millipore, #AB15014, 1:250, Billerica, MA, USA) or anti-GlyR beta (Novis Bio, #NBP2-30683, 1:100, Anseong-si, South Korea) overnight at room temperature. Then, sections were washed and incubated with the secondary antibody (Alexa Fluor 405 donkey anti-rabbit, ThermoFisher, # A48258, 1:100) for 2 h at room temperature. After several rinses with PBS, floating sections were mounted on slides. ProLonged gold antifade mounting media (Thermo-Fisher, #P36930) was used to cover the slides before imaging.

Identification of glycinergic neurons and presympathetic VLM/VMM neurons was performed by using a combination of pAAV-hSyn-DIO-EGFP and PRV-614. First, to identify Cre-expressing glycinergic neurons, the pAAV-hSyn-DIO-EGFP was stereotaxically injected into the VLM (AP: −6.7 mm; ML: ±1.1 mm; DV: −6.3 mm) of GlyT$_2^{Cre}$ mice. After 2–3 weeks, PRV-614 was injected into the kidney of the same mice. After 120 h, the mice were transcardially perfused with 4% paraformaldehyde and 35 μm brain sections were prepared as described above. Then, sections were rinsed in PBS, mounted on slides, dried overnight and covered using ProLong gold antifade mounting media. Fluorescent neurons were visualized with a fluorescence microscope (Olympus BX51) and representative images were acquired with a confocal microscope (Nikon Ti2) using appropriate filters.

## Results

### Enhancing network activity promotes the release of glycine

Patch-clamp recordings were conducted from presympathetic neurons in the VLM/VMM to study GABA and glycine release. First, we determined which inhibitory neurotransmitter generates IPSCs in PRV-labelled neurons by conducting recordings in the presence of the GABA$_A$R blocker, bicuculline. Application of bicuculline (10 μM) reduced the average frequency of sIPSCs from 2.52 ± 2.65 to 1.53 ± 2.13 Hz ($n$ = 25, Wilcoxon matched-pairs signed-rank test, $P$ < 0.0001, Holm–Bonferroni adjusted $P$ = 0.0003; Fig. 1$A$, $B$ and $E$) as well as the amplitude (21.34 ± 8.33 *vs.* 18.22 ± 6.82 pA, $n$ = 24, paired $t$ test, $t_{23}$ = 3.88, $P$ = 0.0008; Holm–Bonferroni adjusted $P$ = 0.0016, not shown). In two additional neurons, application of bicuculline caused an increase of sIPSCs activity probably due to the network's disinhibition; therefore, these cells were not included in the analysis. Consequently, the average inhibitory $I_{phasic}$ current decreased from 0.26 ± 0.35 to 0.14 ± 0.22 pA after bicuculline application ($n$ = 25, Wilcoxon matched-pairs signed-rank test, $P$ < 0.0001, Holm–Bonferroni adjusted $P$ = 0.0002 Fig. 1$H$). These data demonstrate that in presympathetic VLM/VMM neurons, approximately 40% of the total $I_{phasic}$ is bicuculline sensitive (Fig. 1$J$), whereas approximately 60% is bicuculline insensitive. We did not find significant changes in the 10–90% decay time after bicuculline application. The average 10–90% decay time was 17.24 ± 19.79 ms before bicuculline application and 12.46 ± 10.33 ms ($n$ = 24; $P$ = 0.25) after bicuculline application. Before bicuculline application 11 out of the 24 cells showed a single-exponential decay time and 13 cells showed doubled-exponential decay. After bicuculine application we found that five out of the 11 neurons changed their double-exponential decay to single-exponential decay time.

Next, we tested whether an increase in neuronal network activity leads to an increase of bicuculline-insensitive inhibitory postsynaptic currents and whether these currents are inhibited by strychnine,

a GlyR inhibitor, as was observed in rats (Gao et al., 2019). Increased neuronal network activity was achieved by blockade of potassium channels with 4-AP to prevent repolarization of the neurons. In the presence of bicuculline, application of 4-AP (1 mM) increased the frequency of sIPSCs from $0.85 \pm 1.18$ to $7.79 \pm 5.17$ Hz ($n = 9$, Wilcoxon matched-pairs signed-rank test, $P = 0.0039$, Holm-Bonferroni adjusted $P = 0.0078$; Fig. 1*C* and *F*). Furthermore, the amplitude of sIPSCs increased ($17.51 \pm 5.89$ *vs.* $25.93 \pm 12.36$ pA) ($n = 8$, paired *t* test, $t_7 = 2.54$, $P = 0.0384$, Holm–Bonferroni adjusted $P = 0.0384$; not shown) as well as the charge transfer ($0.07 \pm 0.02$ pC *vs.* $0.14 \pm 0.08$ pC) ($n = 8$, paired *t* test, $t_7 = 3.29$, $P = 0.0133$; not shown). Similarly, the application of 4-AP increased $I_{phasic}$ from $0.06 \pm 0.07$ pA to $1.27 \pm 1.20$ pA ($n = 9$, Wilcoxon matched-pairs signed-rank test, $P = 0.0039$, Holm–Bonferroni adjusted $P = 0.0039$; Fig. 1*I*). A strychnine bath was applied to verify that the bicuculline insensitive currents were generated by activation of GlyRs. Figure 1D shows that strychnine application reduced the frequency of sIPSCs to $0.21 \pm 0.13$ Hz ($n = 5$; paired *t* test, $t_4 = 3.721$, $P = 0.0205$, Holm–Bonferroni adjusted $P = 0.0205$; Fig. 1*G*). These results indicate that both GABA and glycine are released and contribute to the generation of sIPSCs in presympathetic VLM/VMM neurons and that enhanced neuronal network activity potentiates the release of glycine.

### Stimulation of GlyT$_2$ fibres in the VLM/VMM triggers release of both GABA and glycine

To assess glycine release from glycinergic inputs, ChR2$^{EYFP}$ was expressed in GlyT$_2$$^{Cre}$ transgenic mice (see Material and Methods). Our experiments revealed that light stimulation (5 ms) of GlyT$_2$$^{ChR2/EYFP}$ fibres evoked IPSCs in 67% of the recorded presympathetic VLM/VMM neurons (Fig. 2). The average amplitude of eIPSCs was $529 \pm 469$ pA (n = 24) and application of bicuculline (10 µM) decreased the amplitude of eIPSCs. In the presence of bicuculline, the amplitude of eIPSCs was $336 \pm 254$ pA ($n = 24$, Dunn's multiple comparisons test, $P = 0.0417$). In two additional cells, application of bicuculline increased the amplitude of eIPSCs, which was probably due to the network's disinhibition; therefore, those cells were not included in the analysis. The application of bicuculline also decreased the charge transfer of light-evoked IPSCs from $5.30 \pm 5.91$ to $2.81 \pm 2.47$ pC ($n = 21$, Dunn's multiple comparisons test, $P = 0.0101$; Fig 2*D*). To verify that the eIPSCs were generated by glycinergic fibre activation, experiments were repeated in wild-type (GlyT$_2$$^{Cre}$-negative) mice, in which light stimulation (5 ms) did not evoke IPSCs in presympathetic VLM/VMM neurons ($n = 10$, data not

shown). Together, these data suggest that GABAergic neurotransmission is responsible only for a portion of inhibitory currents evoked by light stimulation of GlyT$_2$$^{ChR2/EYFP}$ fibres in presympathetic VLM/VMM neurons.

To reveal the contribution of GlyT$_2$ transporter to the modulation of light-evoked neurotransmission in the VLM/VMM, recordings were conducted in the presence of a GlyT$_2$ inhibitor, ORG25543. Application of ORG25543 (10 µM) in the presence of bicuculline significantly decreased the amplitude of light-evoked IPSCs from $336 \pm 254$ to $149 \pm 154$ pA ($n = 24$, Dunn's multiple comparisons test, $P = 0.0311$; Fig. 2*A* and *C*). Consistent with decreased light-evoked IPSC amplitude, the mean charge transfer of light-evoked IPSCs also decreased from $2.81 \pm 2.47$ to $1.04 \pm 0.94$ pC ($n = 21$, Dunn's multiple comparisons test, $P = 0.0021$; Fig. 2*D*).

Strychnine was applied in the presence of bicuculline and ORG25543 (Fig. 2*A* and *C*) to evaluate whether the remaining light-evoked IPSCs were generated by activation of GlyRs. Strychnine blocked the remaining light-evoked IPSCs in presympathetic VLM/VMM neurons ($n = 24$, $3.16 \pm 5.62$ pA). These data demonstrate that approximately 70% of the light-evoked currents are bicuculline insensitive and GlyR blockade inhibited the remaining amplitude of light-evoked IPSCs.

### Glycinergic inputs to presympathetic VLM/VMM neurons are monosynaptic

Monosynaptic connections refer to the presence of direct connections between two neurons. To demonstrate mono-synaptic connections between GlyT$_2$$^{ChR2/EYFP}$-expressing neurons and presympathetic VLM/VMM neurons, first we used TTX to block voltage-gated sodium channels activated by ChR2-dependent membrane depolarization. Then, 4-AP (1 mM) was applied to block voltage-dependent potassium channels to enhance the direct depolarization of monosynaptic inputs by ChR2, thus restoring eIPSCs (Petreanu et al., 2009; Shu et al., 2007). Application of TTX (1 µM) abolished the light-evoked IPSCs ($164 \pm 115$ *vs.* 0 pA, $n = 11$, Dunn's multiple comparisons test, $P < 0.0001$; Fig. 3*A* and *B*), whereas subsequent application of 4-AP partially restored light-evoked IPSCs ($58.15 \pm 73.94$ pA, $n = 11$, Dunn's multiple comparisons test, $P = 0.0315$; Fig. 3*A* and *B*). The recovered response displayed a delay in light-evoked IPSC activation from $7.11 \pm 0.92$ to $18.13 \pm 7.18$ ms ($n = 11$, paired *t* test, $t_{10} = 5.43$, $P = 0.0003$; Fig. 3*C*) compared with control. This increased time delay is attributed to the slow kinetics of ChR2-dependent depolarization of neurons with the blockade of voltage-gated sodium channels (Petreanu et al., 2009). Together, our data

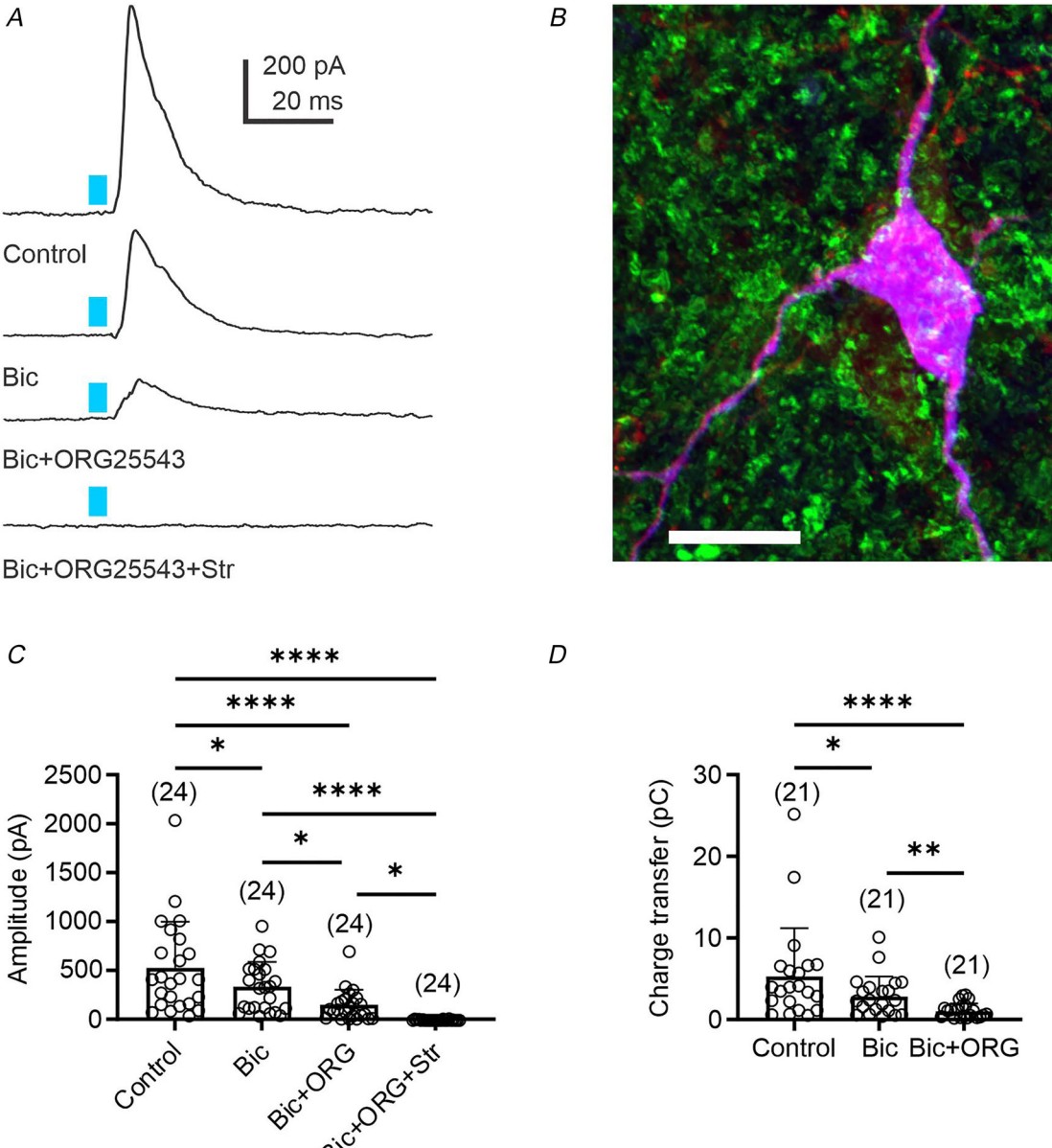

**Figure 2. Light stimulation of glycinergic fibres in the VLM/VMM results in release of both GABA and glycine**

*A*, light-evoked IPSCs recorded from a presympathetic VLM/VMM neuron ($E = -10$ mV) after a 5 ms light stimulation (Control), followed by bicuculline (10 µM), ORG25543 (10 µM) and strychnine (1 µM). Application of bicuculline revealed the glycinergic component of light-evoked IPSCs. *B*, example of a recorded presympathetic neuron. Magenta represents the combination of red fluorescence (PRV-614) and blue fluorescence (avidin–biotin AMCA fluorescent reaction) illustrating that the recording was conducted from a presympathetic neuron. Green represents GlyT$_2$$^{ChR2/EYFP}$-expressing fibres. Scale bar = 10 µm. *C*, summary data demonstrating the amplitude of light-evoked IPSCs with and without bicuculline, following co-application of bicuculline and ORG25543, and strychnine. Co-application of bicuculline and ORG25543 revealed that GlyT$_2$ controls the glycine content of the vesicles. *D*, summary of bicuculline and ORG25543 effects on charge transfer. Application of bicuculline and co-application of bicuculline and ORG25543 decreased the amplitude of light-evoked IPSCs, resulting in a charge transfer decrease. Numbers of replications are shown in parentheses. Bar graphs represent the mean ± SD; open circles represent individual data points. Statistical analyses were performed using Dunn's multiple comparisons test (*C*, *D*). *$P < 0.05$, **$P < 0.01$, ****$P < 0.0001$.

suggest that presympathetic VLM/VMM neurons receive monosynaptic inputs from glycinergic neurons.

### Short-term plasticity of GlyT$_2$-expressing neurons

Many synapses exhibit a variety of short-term depression forms. Synaptic depression that occurs over milliseconds and seconds is caused by changes in the amount of transmitter release from presynaptic terminals (Jiang & Abrams, 1998; Regehr, 2012). Repeated stimulation of presynaptic neurons results in reduced evoked current amplitude in the postsynaptic cells. The amplitude of evoked current is most depressed at short interpulse interval (IPI), and the amplitude recovers as IPI increases (Doyle & Andresen, 2001; Zucker & Regehr, 2002). In our study, presynaptic release probability changes were examined after a train of stimuli, which were calculated as the amplitude of peak n ($P_n$) divided by the amplitude of peak 1 ($P_n/P_1$).

Under control conditions, at a holding potential of $-10$ mV, a train of four light stimuli was used to trigger eIPSCs in presympathetic VLM/VMM neurons (Fig. 4). Although most of the eIPSCs had a single peak, in some cases light stimulation evoked multiple peaks (data not shown). As shown on Fig. 4, with a 50 ms IPI, a large degree of depression was observed after the first stimulus; however, the amplitude of the second, third and fourth light-evoked IPSCs did not differ. The mean ratio for $P_2/P_1$ was $0.32 \pm 0.22$ (range 0–0.61, $n = 10$), $P_3/P_1$: $0.22 \pm 0.16$ (range 0–0.48, $n = 10$) and $P_4/P_1$: $0.25 \pm 0.21$ (range 0–0.55, $n = 10$) showing no significant difference between $P_2/P_1$, $P_3/P_1$ and $P_4/P_1$. The 500 ms IPI resulted in a

depression of eIPSCs; however, the mean ratio increased to $P_2/P_1$: $0.50 \pm 0.0.23$ (range 0.09–0.69, $n = 10$), $P_3/P_1$: $0.53 \pm 0.18$ (range 0.06–0.65, $n = 10$) and $P_4/P_1$: $0.49 \pm 0.19$ (range 0.10–0.75, $n = 10$), (Fig. 4A and B) compared to the mean ratio seen with 50 ms IPI. As the IPI increased to 2000 ms the depression previously observed after the first stimulus was diminished (Fig. 4A and B). The mean ratio for $P_2/P_1$ was $0.88 \pm 0.18$ (range 0.69–1.32, $n = 10$), $P_3/P_1$: $0.87 \pm 0.49$ (range 0.39–2.21, $n = 10$) and $P_4/P_1$: $0.84 \pm 0.15$ (range 0.63–1.19, $n = 10$). Despite significant changes in the mean ratio between the three IPI curves, there were no significant differences between $P_2/P_1$, $P_3/P_1$ and $P_4/P_1$ within the same IPI (Fig. 4B). Therefore, our data demonstrate a IPI-dependent decrease of neurotransmitter release. It is possible that a shorter interval promotes depletion of the readily releasable pool of vesicles and/or insufficient time for vesicle recruitment.

Glycinergic neurons in the brainstem and spinal cord express GlyT$_2$, which mediates uptake of glycine (Liu et al., 1993; Rousseau et al., 2008; Zafra et al., 1995). Here, we examined whether GlyT$_2$ controls the dynamic of glycine release. Application of bicuculline significantly reduced the amplitude of first light-evoked IPSCs to approximately 68% (reduction ranged from 31 to 104%; Tukey's multiple comparisons test; $P = 0.0016$; $n = 27$). Additionally, peaks 2, 3 and 4 in the presence of bicuculline exhibited significant reductions compared to control traces (Tukey's multiple comparisons test; $P < 0.05$; $n = 27$). Despite the significant decrease of the first eIPSC amplitude after bicuculline application, the paired-pulse ratio (PPR) did not change and the amplitudes of evoked IPSC peaks 2, 3 and 4 also remained unchanged (Fig. 5A–C). These data indicate that at 50 ms IPI, the light-evoked IPSCs

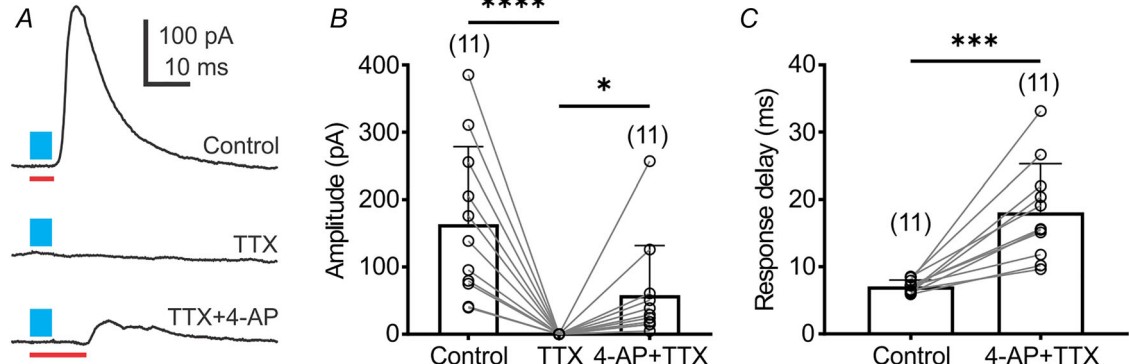

**Figure 3. Monosynaptic glycinergic inputs to presympathetic neurons in the VLM/VMM**
*A*, light-evoked IPSCs recorded from a presympathetic VLM/VMM neuron ($E = -10$ mV) after a 5 ms light pulse (Control), followed by TTX application (1 μM), then co-application of TTX and 4-AP (TTX, 1 μM + 4-AP, 1 mM). Application of TTX prevented the eIPSCs, whereas co-application of TTX and 4-AP reinstated the light-evoked IPSC with a response delay. The blue bar indicates light stimulation. *B*, group data showing the amplitude of light-evoked IPSCs. *C*, group data demonstrating an increase in the delay time of light-evoked IPSCs. Asterisks indicate significant differences between groups. Numbers of replicates is shown in parentheses. Bar graphs represent the mean ± SD; open circles represent individual data points. Statistical analyses were performed using Dunn's multiple comparisons test (*B*) and paired *t* test (*C*). *$P < 0.05$, ***$P < 0.001$, ****$P < 0.0001$.

generated by glycine release showed no changes in presynaptic release probability (Fig. 5C).

Next, we provide insight into whether blockade of $GlyT_2$ decreases presynaptic release probability. Trains of light-evoked IPSCs recorded before and after application of ORG25543 revealed no changes in PPR. Application of ORG25543 significantly reduced the amplitude of the first light-evoked IPSCs to ∼54% (reduction ranged from 15 to 108%; Tukey's multiple comparisons test, $P = 0.0040$; $n = 16$, Fig. 5D–F). Peaks 2, 3 and 4 in the presence of ORG25543 also exhibited significant reductions compared to control traces (Tukey's multiple comparisons test; $P < 0.05$; $n = 16$), although the PPR was not different before and after ORG25543 application. These data indicate that at 50 ms IPI, blockade of $GlyT_2$ showed no changes in presynaptic release probability of GABA (Fig. 5F).

In the spinal cord GABA and glycine may substitute for one another when their relative concentration is changed (e.g. during blockade of $GlyT_2$) (Rousseau et al., 2008; Wojcik et al., 2006). Here, we found that in the presence of bicuculline, subsequent application of ORG25543 decreased both the amplitude of light-evoked IPSCs and PPR (Fig. 5G–I). Application of ORG25543 in the presence of bicuculline significantly reduced the amplitude of the first light-evoked IPSCs to ∼39% (reduction ranged

from 13 to 77%), as well as peaks 2, 3 and 4 (Tukey's multiple comparisons test; $P < 0.05$; $n = 8$). These data suggest that $GlyT_2^{ChR2/EYFP}$-expressing synapses displayed a decrease in presynaptic glycine release probability after ORG25543 application. It is possible that the decreased amplitude of light-evoked IPSCs is caused by a decrease in residual glycine levels in the synaptic vesicles and/or in the number of glycinergic vesicles. The large degree of depression suggests that the residual level of glycine decreased in the synaptic vesicles. Consequently, our data demonstrate that $GlyT_2$ plays an essential role in recovering glycinergic neurotransmission in the VLM/VMM.

## Identification of glycinergic inputs to presympathetic VLM/VMM neurons

To identify the location of glycinergic neurons with projections to presympathetic VLM/VMM neurons, pAAV-hSyn-DIO-EGFP was injected into the left VLM of $GlyT_2^{Cre}$ mice (Fig. 6). EGFP-positive fibres were found in many brain areas including the lateral preoptic area, lateral hypothalamus, thalamus, red nucleus, superior colliculus, lateral lemniscus, inferior olive and reticular nucleus (from the anterior formation to the medullary part). On

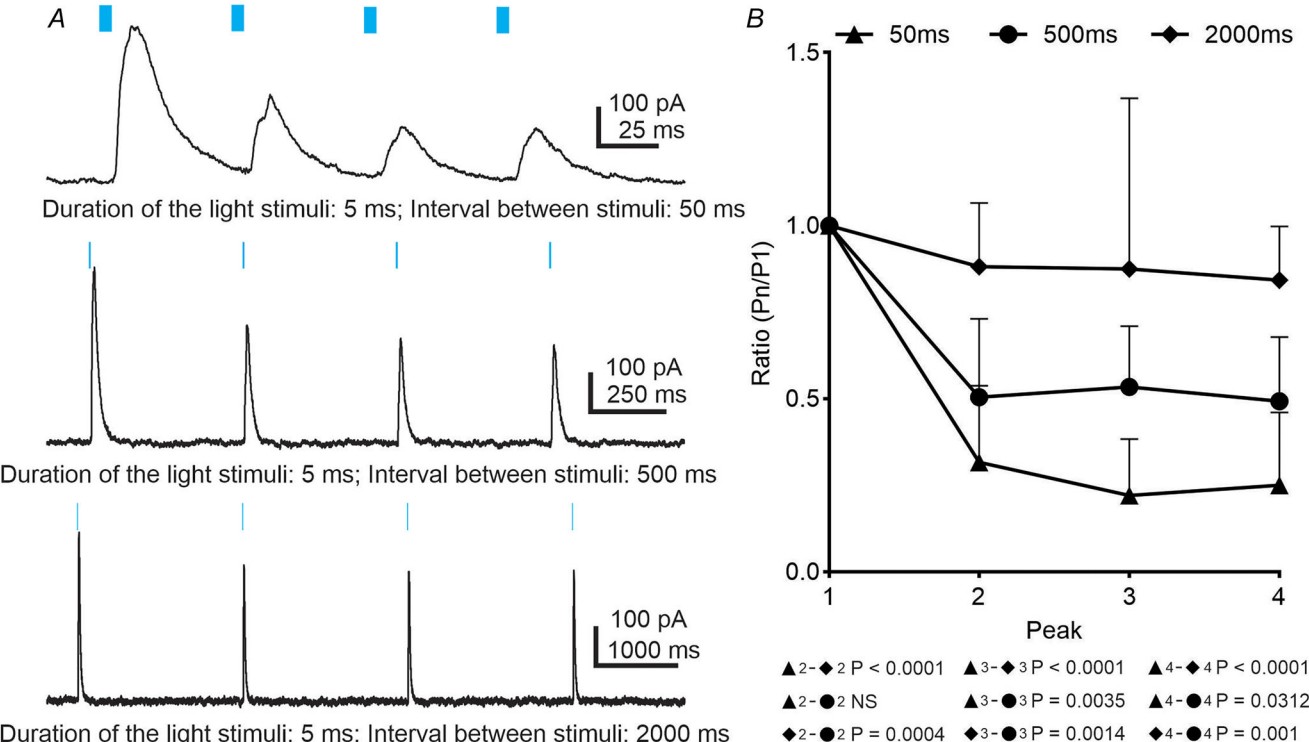

**Figure 4. Sustainability of neurotransmission during train of stimuli**
*A*, train of light-evoked IPSCs with different interpulse intervals (IPI) was recorded at −10 mV from presympathetic neurons. *B*, frequency-dependent changes in amplitude of light-evoked IPSCs at different IPI. Statistical analyses were performed using Turkey's multiple comparisons test.

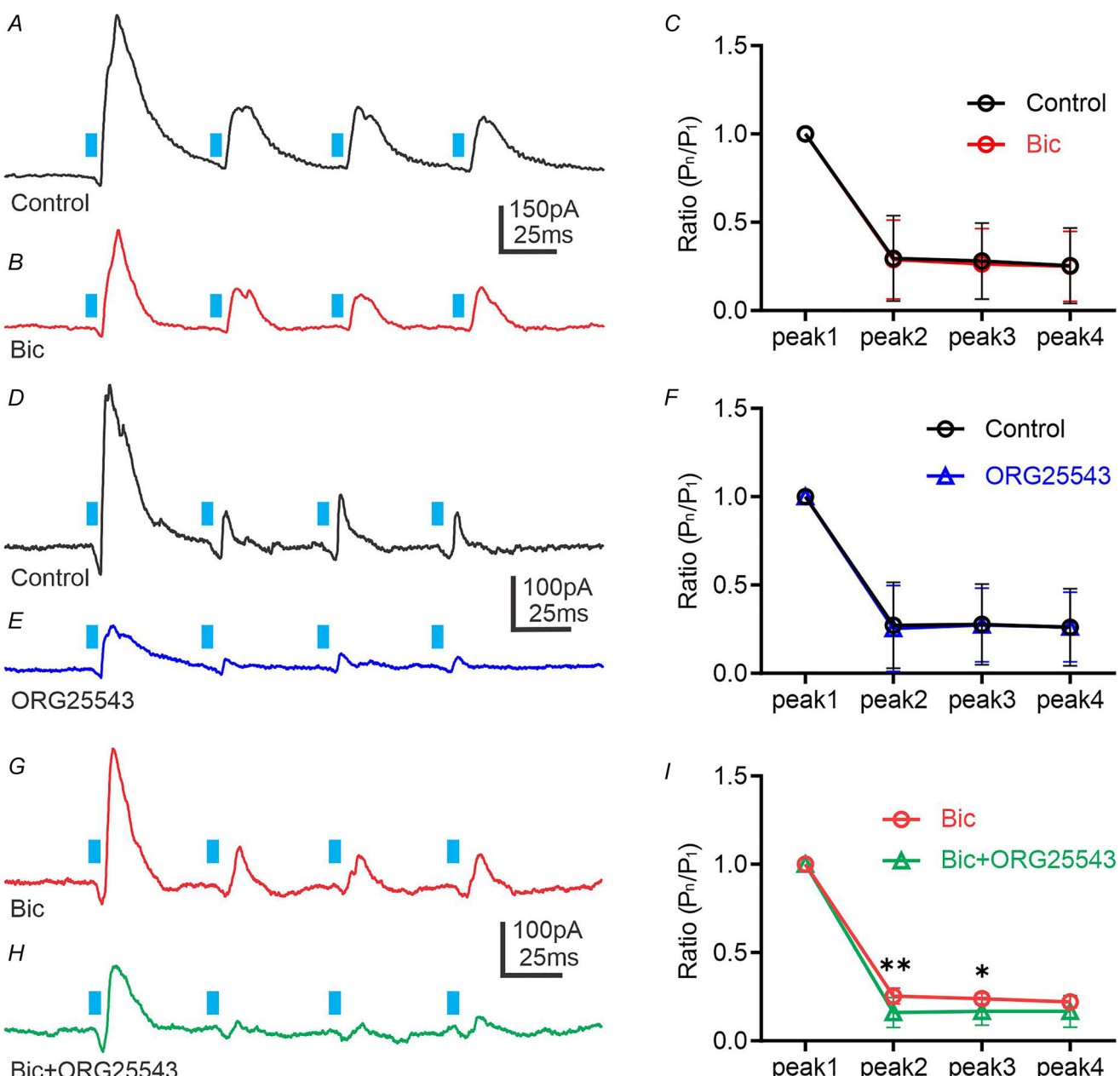

**Figure 5. Inhibition of GlyT$_2$ reduces glycine release**

*A* and *B*, train of light-evoked glycinergic IPSCs was recorded at −10 mV from PRV-labelled neurons before (*A*) and after application of bicuculline (10 μM, *B*). The duration of light stimuli was 5 ms. The interpulse interval (IPI) between the stimuli was 50 ms. *C*, summary data representing the ratio between the amplitude of peak n (P$_n$) divided by the peak 1 amplitude (P$_1$). Before bicuculline application, the mean ratio of P$_2$/P$_1$ was $0.30 \pm 0.24$ (range 0–0.79, *n* = 27), P$_3$/P$_1$: $0.28 \pm 0.22$ (range 0–0.75, *n* = 27) and P$_4$/P$_1$: $0.25 \pm 0.21$ (range 0–0.72, *n* = 27). The mean ratio of responses after bicuculline application was P$_2$/P$_1$: $0.29 \pm 0.22$ (range 0–0.79, *n* = 27), P$_3$/P$_1$: $0.26 \pm 0.20$ (range 0–0.75, *n* = 27) and P$_4$/P$_1$: $0.25 \pm 0.20$ (range 0–0.73, *n* = 27). *D* and *E*, train of light-evoked IPSCs recorded at −10 mV from PRV-labelled neurons before (*D*) and after application of ORG25543 (10 μM, *E*). *F*, summary data representing the ratio between the amplitude of peak n (P$_n$) divided by the amplitude of peak 1 (P$_1$). The mean ratio in control was P$_2$/P$_1$: $0.27 \pm 0.24$ (range 0.04–0.72, *n* = 16), P$_3$/P$_1$: $0.28 \pm 0.23$ (range 0.05–0.73, *n* = 16) and P$_4$/P$_1$: $0.26 \pm 0.22$ (range 0.02–0.71, *n* = 16). The mean ratio of responses after ORG25543 application was P$_2$/P$_1$: $0.25 \pm 0.24$ (range 0.03–0.80, *n* = 16), P$_3$/P$_1$: $0.27 \pm 0.21$ (range 0.08–0.71, *n* = 16) and P$_4$/P$_1$: $0.26 \pm 0.20$ (range 0.004–0.64, *n* = 16). *G* and *H*, train of light-evoked glycinergic IPSCs recorded at −10 mV from PRV-labelled neurons before (*G*) and after application of ORG25543 (10 μM) in bicuculline (10 μM) (*H*). *I*, summary data representing the ratio between the amplitude of peak n (P$_n$) divided by the amplitude of peak 1 (P$_1$). The mean ratio in the presence of bicuculline was P$_2$/P$_1$: $0.25 \pm 0.05$ (range 0.21–0.32, *n* = 8), P$_3$/P$_1$: $0.24 \pm 0.03$ (range 0.20–0.28, *n* = 8) and P$_4$/P$_1$: $0.22 \pm 0.03$ (range 0.18–0.26, *n* = 8). The mean ratio of responses after

ORG25543 application was $P_2/P_1$: $0.16 \pm 0.08$ (range 0.04–0.32, $n = 8$), $P_3/P_1$: $0.17 \pm 0.08$ (range 0.06–0.30, $n = 8$) and $P_4/P_1$: $0.17 \pm 0.09$ (range 0.07–0.35, $n = 8$). Statistical analyses were performed using Turkey's multiple comparisons test and Sidak's multiple comparisons test (*C*, *F*, *I*).

the other hand, EGFP-positive cell bodies were observed in the area surrounding the VLM/VMM, with abundant expression in the RVLM, lateral paragigantocellular nucleus and Bötzinger complex. These data show that glycinergic neurons with projections to presympathetic VLM/VMM neurons are mainly located in the ventral brainstem.

Next, injection of retrograde AAV (pAAV-hSyn-DIO-EGFP) into the VLM was combined with PRV-614 inoculation of the kidney to identify the location of glycinergic neurons with projections to presympathetic VLM/VMM neurons. Our electrophysiological recordings from presympathetic VLM/VMM neurons were conducted 96 h after kidney inoculation (Gao & Derbenev, 2013; Gao et al., 2019); therefore, an additional 24 h was needed to determine the location of the next order of neurons. Double labelled cells expressing GFP (glycinergic) and RFP (presympathetic) were observed in the VLM/VMM, supporting our findings that presympathetic VLM/VMM neurons receive projections from glycinergic neurons located in the neighbouring area. These results also revealed that a subset of higher order kidney-related neurons express $GlyT_2$ in the ventral brainstem (Fig 6).

## Differential distribution of GlyR subunits in the ventral brainstem

In the CNS, GlyRs are pentameric oligomers composed of $\alpha$ and $\beta$ subunits, although the stoichiometry of the subunits is debated (Grudzinska et al., 2005; Lynch, 2004, 2009). A homomeric or heteromeric combination of $\alpha$ subunits determines the ligand-binding domain; however, all cell surface membrane GlyRs express $\beta$ subunits as they are responsible for clustering GlyRs at the post-synaptic membrane and anchoring GlyRs to the cytoplasmic protein via gephyrin (Meyer et al., 1995). We used immunofluorescence staining to determine the subunit composition of GlyRs in presympathetic VLM/VMM neurons (Fig. 7). Previous data showed that $\alpha 2$ and $\alpha 4$ subunits are primarily important for development (Ceder et al., 2024; Darwish et al., 2023; Nishizono et al., 2020), so we focused on $\alpha 1$, $\alpha 3$ and $\beta$. GlyRs were labelled with primary antibody recognizing an extracellular epitope of the $\alpha 1$ subunit, as described previously (Pfeiffer et al., 1984), and scattered distribution of $\alpha 1$ immunofluorescence was observed throughout the RVLM (Fig. 7*C* and *D*). Then, primary antibody recognizing $\alpha 3$ subunits was used and, unlike the labelling for $\alpha 1$ subunits,

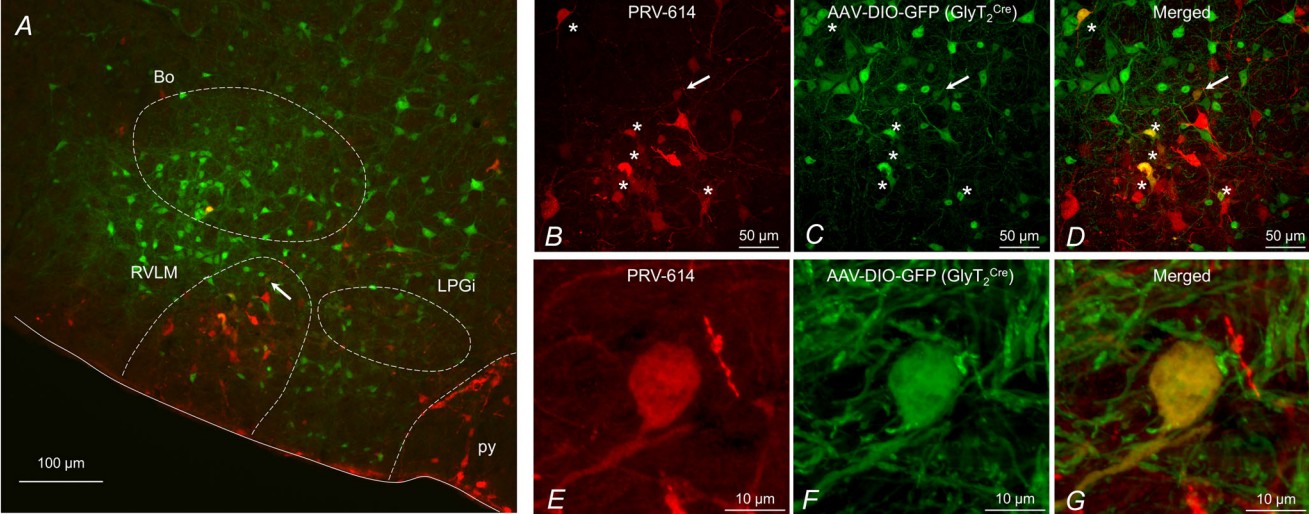

**Figure 6. Location of presympathetic glycinergic neurons in the VLM/VMM**
*A*, an epifluorescence microscope image (10× magnification) of a brainstem section (35 μm) demonstrating the location of $GlyT_2$-expressing neurons (EGFP) and presympathetic neurons (RFP). Double-labelled neurons were found in the VLM/VMM and other areas indicating glycinergic neurons projecting to presympathetic neurons in the VLM/VMM. Arrow indicates a double-labelled neuron. *B–D*, 40× magnification confocal image of the RVLM area of the same section shown in *A* displaying presympathetic neurons (RFP) and $GlyT_2$-expressing neurons (EGFP). Stars indicate double-labelled neurons. Arrow indicates same double-labelled neuron shown on *A*. *E–G*, 60× magnification confocal image of a double-labelled neuron in the VLM/VMM labelled with an arrow on *A–D*. RVLM, rostral ventrolateral medulla; Bo, Bötzinger complex; LPGi, lateral paragigantocellular nucleus; py, pyramidal tractus.

we found that $\alpha 3$ subunits displayed diffuse distribution within the soma of presympathetic VLM/VMM neurons (Fig. 7*G* and *H*). Similarly, $\beta$ subunit immunofluorescence was observed in the VLM/VMM (Fig 7*K* and *L*). Our data demonstrate differential distribution of GlyRs containing $\alpha 1$ and $\alpha 3$ subunits in the VLM/VMM of mice, suggesting that $\alpha 1$ GlyRs are located on dendrites of presympathetic neurons in the VLM/VMM, whereas GlyRs $\alpha 3$ are located on the soma.

## Discussion

Our study demonstrates that in the ventral brainstem, GlyT2-expressing neurons rely on both $GABA_A$ and glycine receptors to generate IPSCs. We found that the sIPSCs recorded from presympathetic neurons in the ventral brainstem are generated by presynaptic release of GABA and glycine and increased network activity led to increased glycine release. Light stimulation of glycinergic fibres triggered release of both GABA and glycine. Our

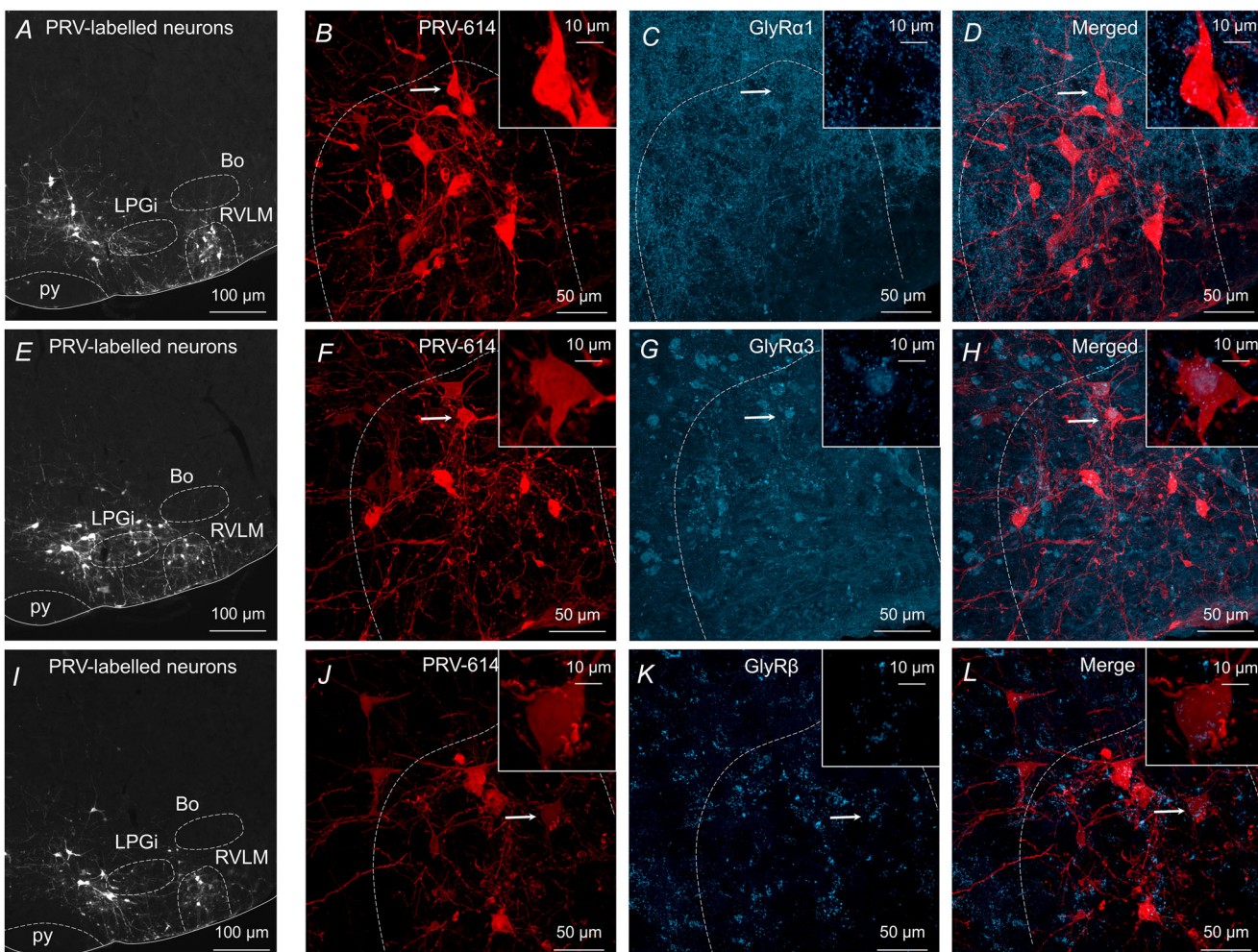

**Figure 7. GlyR subtypes in VLM/VMM**

*A*, low-magnification image showing presympathetic neurons in the rostral brainstem 96 h after inoculation of the left kidney. *B*, enlarged confocal image of the RVLM shown in *A*. The boxed area in the right upper corner is an enlarged image of the presympathetic neuron indicated by the arrow. *C*, confocal image of GlyR$\alpha 1$ immunoreactive puncta distributed throughout the RVLM. The enlarged boxed area illustrates $\alpha 1$ immunopositive puncta around a presympathetic VLM/VMM neuron. *D*, merged image showing presympathetic neurons and GlyR$\alpha 1$ immunoreactive puncta in the RVLM. *E*, low-magnification image of presympathetic neurons in the ventral brainstem. *F*, enlarged image of the RVLM shown in *E*. The boxed area in the right upper corner is an enlarged image of the presympathetic neuron indicated by the arrow. *G*, confocal image of GlyR$\alpha 3$ immunoreactive staining in the RVLM. *H*, merged image illustrating GlyR$\alpha 3$ immunopositive presympathetic neurons in the RVLM. *I*, low-magnification image of presympathetic neurons in the ventral brainstem. *J*, enlarged confocal image of the RVLM shown in *I*. The boxed area in the right upper corner is an enlarged image of the presympathetic neuron indicated by the arrow. *K*, confocal image of GlyR$\beta$ immunoreactive puncta in the RVLM. *L*, merged image illustrating GlyR$\beta$ immunopositivity in the RVLM.

anatomical study identified the location of glycinergic neurons in the ventral brainstem, and we found that GlyRs expressed in presympathetic VLM/VMM neurons are formed by $\alpha 1$, $\alpha 3$ and $\beta$ subunits. Together, our findings demonstrate novel release patterns of GABA and glycine from glycinergic fibres in the VLM/VMM and suggest that both inhibitory neurotransmissions should be considered when the functionality of presympathetic neurons and sympathetic output are discussed.

## IPSCs in presympathetic VLM/VMM neurons generated by GABA and glycine

Presympathetic neurons in the VLM/VMM integrate descending inputs from multiple brain areas and represent a major output to sympathetic preganglionic neurons located in the intermediolateral cell column of the spinal cord (Dampney, 1994; Dampney et al., 2003; Gao et al., 2019; Guyenet, 2006; Zsombok et al., 2024). The excitability of presympathetic neurons in the VLM/VMM depends on multiple factors, including the intrinsic properties of neurons, the balance between the activity of inhibitory and excitatory synaptic inputs, and paracrine and endocrine signalling. In the RVLM, GABA was identified as a primary inhibitory neurotransmitter that mediated most IPSCs (Gao & Derbenev, 2013; Gao et al., 2019; Hayar et al., 1996). On the other hand, spontaneous IPSCs generated by activation of GlyR were also identified in the RVLM (Gao et al., 2019). Moreover, anatomical studies revealed that synaptic terminals and neurons in the VLM/VMM, including neurons in the Bötzinger area, co-express GABA and glycine (Llewellyn-Smith et al., 2001; Schreihofer et al., 1999); however, the mechanism of glycine release remains elusive. Previously we showed that blockade of GlyRs in the rat RVLM shortened the recovery time of renal sympathetic nerve activity after an increase of blood pressure (Gao et al., 2019), suggesting that GABA controls threshold excitability, whereas glycine increases the strength of inhibition.

In this study we have demonstrated that GlyT2-expressing neurons release both GABA and glycine to inhibit presympathetic neurons in the VLM/VMM. Previous studies reported that 10–30 μM bicuculline effectively blocks GABAergic IPSCs in the brainstem, hypothalamus and spinal cord (Gao & Derbenev, 2013; Gao & Smith, 2010; Gao et al., 2012, 2019; Jiang et al., 2013; Takazawa & MacDermott, 2010); therefore, bicuculline was used to block GABAergic sIPSCs and reveal the glycinergic sIPSCs. In our study male and female mice were used; however, our analysis of spontaneous IPSCs did not reveal a sex-dependent difference in frequency and amplitude; thus, the pooled data were presented. Recordings of sIPSCs from presympathetic neurons in the presence of bicuculline determined that ∼60%

of the sIPSCs were bicuculline insensitive. Recordings conducted in the presence of strychnine confirmed that the remaining sIPSCs were generated by GlyR, which suggests that glycinergic sIPSCs generate about half of the spontaneous inhibitory postsynaptic currents. In addition, our data showed that network excitability has a prominent effect on glycine release, because in the presence of a GABA$_A$R blocker, increased network activity due to 4-AP application led to increased frequency of sIPSCs. Together, these results strongly suggest that in presympathetic VLM/VMM neurons sIPSCs are generated by both GABA and glycine.

GABA and glycine co-release was reported (Jonas et al., 1998; Lu et al., 2008) leading to mixed IPSCs with a characteristic biphasic time course (Aubrey & Supplisson, 2018). This distinctive shape arises from different kinetics of the currents mediated by GABA$_A$R and GlyR. We assessed IPSC decay time before and after bicuculline application. Unfortunately, we could not clearly separate GABAergic and glycinergic IPSCs based on their kinetics. It is possible that dendritic filtering affects the decay time. The IPSCs generated at the distal dendrite appear slow and smaller at the soma due to electrical filtering and cable properties in contrast to IPSCs generated closer to the soma, which can prevent clear separation of GABAergic from glycinergic sIPSCs in our study (Aubrey & Supplisson, 2018).

## Monosynaptic glycinergic input releases both GABA and glycine

Glycinergic neurons were identified within the vestibular and cardio-respiratory brainstem areas (Hirrlinger et al., 2019; Llewellyn-Smith et al., 2001; McMenamin et al., 2016; Schreihofer et al., 1999; Stornetta et al., 2004; Zafra et al., 1995). Previous work reported that both GABA and glycine were found in the same synaptic terminals or cell bodies throughout the medulla oblongata (Llewellyn-Smith et al., 2001; Stornetta et al., 2004). Moreover, single-cell transcriptomic analysis identified a neuronal cluster in the midbrain with GABA–glycine dual transmitters (Yao et al., 2023). Our previous study revealed that mixed sIPSCs were generated by GABA and glycine in the rat RVLM (Gao et al., 2019) and our current study demonstrates release of both GABA and glycine from glycinergic fibres in the mouse brainstem. We showed that light stimulation of GlyT$_2$[ChR2/EYFP] fibres evoked IPSCs, which are generated by presynaptic release of both GABA and glycine. Application of a GABA$_A$R blocker reduced the amplitude of light-evoked IPSCs and a cocktail of bicuculline and strychnine blocked the remaining evoked IPSCs. This finding suggests that the IPSCs evoked by the light stimulation of GlyT$_2$[ChR2/EYFP] fibres are composed of GABA and glycine released

within the same or near the same time interval. In addition, our electrophysiological data demonstrated that the $GlyT_2^{ChR2/EYFP}$ synaptic inputs are monosynaptic, because the light-evoked IPSCs were blocked by TTX and recovered after application of 4-AP. Together, our data show that the inhibitory postsynaptic currents, both sIPSCs and light-evoked IPSCs, are generated by GABA and glycine with monosynaptic connections between glycinergic neurons and presympathetic VLM/VMM neurons.

Presympathetic inhibitory neurons in the VLM/VMM display GABAergic and glycinergic phenotypes and express VGAT and $GlyT_2$ (Muller et al., 2020; Stornetta et al., 2004). $GlyT_2$ supplies the glycine in nerve terminals to promote VGAT-mediated vesicular filling (Aubrey et al., 2007). Our findings indicate that the blockade of $GlyT_2$ nearly eliminates the release of glycine from the $GlyT_2^{ChR2/EYFP}$-expressing neurons during single light stimulation and train stimulation (Figs 2 and 5). Interestingly, during train stimulation, only the amplitude of the first evoked IPSCs remains relatively large, probably due to residual glycine recovery in the synapse or additional mechanisms supplying glycine in the neurons. Therefore, our data confirmed that blockade of $GlyT_2$ reduces glycine reuptake followed by decrease of glycine release from inhibitory glycinergic synapses. This indicates an essential role for $GlyT_2$ in glycine recycling and glycinergic neurotransmission in the ventral brainstem.

### Presympathetic glycinergic neurons and postsynaptic expression of GlyR

Our electrophysiological data demonstrated that $GlyT_2$-expressing neurons directly project to presympathetic neurons identified by PRV 96 h after inoculation, and immunostaining that confirmed the presence of GlyRs in these neurons. Furthermore, pAAV-hSyn-DIO-EGFP injection into the VLM of $GlyT_2^{Cre}$ mice revealed the location of glycinergic neurons in the ventral brainstem. We found abundant expression of glycinergic neurons throughout the VLM/VMM, lateral paragigantocellular nucleus and Bötzinger complex. Glycinergic fibres were identified in many brain areas including the lateral preoptic area, lateral hypothalamus, red nucleus and the reticular formation; however, the cell bodies were located in areas surrounding the VLM/VMM. In addition, we observed that some glycinergic neurons co-localized with PRV-expressing neurons 120 h after inoculation of the left kidney, which further suggests that presympathetic neurons labelled at 96 h receive inputs from glycinergic neurons. Identification of glycinergic neurons in the ventral brainstem is consistent with the view of Stornetta

et al. (2004) that the rostral brainstem contains many GABAergic and glycinergic presympathetic neurons that can release both neurotransmitters. Earlier studies showed that portions of sympathetic preganglionic neurons displayed GABAergic and glycinergic monosynaptic inhibitory postsynaptic potentials evoked by RVLM neuron stimulation (Deuchars et al., 1997; Dun & Mo, 1989). Our finding that presympathetic neurons receive inputs from neurons immediate to the RVLM is also consistent with previous anatomical experiments showing that 50% of RVLM neuron inputs are found within the VLM/VMM (Dempsey et al., 2017). Together, these data support that presympathetic neurons in the VLM/VMM receive glycinergic inputs from local neuronal connections.

### Physiological significance

Evidence suggests that co-release of GABA and glycine may determine the strength and timing of inhibition (Russier et al., 2002). In the auditory brainstem nucleus, GABA binds to GlyRs as a co-agonist to accelerate the kinetics of IPSCs (Lu et al., 2008; Moore & Trussell, 2017). In addition to postsynaptic receptor activation, these neurotransmitters can retrogradely act on presynaptic receptors and affect vesicular packaging of neurotransmitters (Hnasko & Edwards, 2012). It has also been reported that the release of glycine can modulate NMDA receptors (Johnson & Ascher, 1987). In addition, glycinergic neurotransmission was suggested to play a critical role in the coordination of respiratory motor outflow and the formation of euponoeic-like breathing pattern (Dutschmann & Paton, 2002; Paton & Richter, 1995; Shao & Feldman, 1997). Moreover, our previous study showed that in the rat RVLM, blockade of GlyRs decreased the time course of baroreflex-mediated sympathoinhibition (Gao et al., 2019). This may suggest that GABA controls threshold excitability, whereas glycine increases the strength of inhibition during increased synaptic activity. Our current study revealed that glycinergic inputs ($GlyT_2$ expressing) release both GABA and glycine. The most likely explanation for the dual-component nature of the sIPSCs is the quantal release of GABA and glycine from a synaptic vesicle located in the glycinergic interneurons. It is possible that the dual component reflects GABA and glycine co-release from the same neuron and, in our opinion, this should be considered when the underlying mechanisms of sympathetic activity are discussed.

It has been shown that $GlyT_2$ is in the axon terminals of glycinergic neurons and largely contributes to glycine uptake (Aubrey et al., 2007; Gomeza et al., 2003; Rees et al., 2006; Wojcik et al., 2006). Earlier studies identified GABAergic and glycinergic presympathetic neurons in

the rostral brainstem based on the presence of GAD-67 and GlyT$_2$ mRNA expression (Morgado-Valle et al., 2010; Stornetta et al., 2004). According to our previous electrophysiological data, bath application of GABA or glycine increases membrane conductance, hyperpolarizes the neurons and decreases their firing rate (Gao & Derbenev, 2013; Gao et al., 2019). Several studies demonstrated expression of GABA$_A$R subtype in the RVLM; however, to our knowledge, no studies have shown expression of GlyRs. Here, in conjunction with our electrophysiological study, we demonstrated differential distribution of GlyRs containing $\alpha$1, $\alpha$3 and $\beta$ subunits in the VLM/VMM of mice. Our immunofluorescence staining identified two differential patterns of GlyR expression. GlyR $\alpha$1 immunoreactive puncta were present along neuronal processes of presympathetic VLM/VMM neurons, whereas GlyR $\alpha$3 immunoreactivity was observed at the soma. It is generally accepted that most IPSCs in adult spinal cord, brainstem and retina are generated by GlyR $\alpha$1 (Lynch, 2004). On the other hand, there is limited knowledge about the functionality of GlyR $\alpha$3 in the brainstem. It has been reported that expression of the GlyR $\alpha$3 increases with the age (Lynch, 2004) and knockout of GlyR $\alpha$3 decreased pain evoked by chronic peripheral inflammation (Harvey et al., 2004). Widespread expression of $\beta$ subunits is known in the CNS as they are an important part of the heteromeric GlyRs and play role in synapse stabilization and ligand binding (Grudzinska et al., 2005). Accordingly, $\beta$ subunits were detected in the ventral brainstem; however, further studies are required to determine the contribution of GlyR subunits to the regulation of sympathetic output.

### Methodological consideration

Our study shows that light stimulation of glycinergic inputs in the VLM/VMM results in both GABA and glycine release. On the other hand, we do not know whether the release of neurotransmitters is from the same vesicles or from separate pools of vesicles (Vaaga et al., 2014). Based on our electrophysiological findings, it is likely that GABA and glycine are released from the same input; however, our study does not provide direct evidence for the co-release of these neurotransmitters and future studies are needed to investigate this scenario.

Our approaches have certain technical limitations. We used PRV to identify presympathetic neurons in the VLM/VMM. The spread of PRV is strictly retrograde and transsynaptic (Strack & Loewy, 1990), and PRV was used to identify neurons as described previously (Derbenev et al., 2004; Gao & Derbenev, 2013; Gao et al., 2019; Jiang et al., 2013). The number and location of infected neurons depend on the post-inoculation survival time. Therefore, infection was optimized to minimize the likelihood of long-term infection effects and reduce the possibility of labelling neurons afferent to our presympathetic RVLM neurons, as we previously reported (Gao & Derbenev, 2013; Gao et al., 2019). Our data suggest that presympathetic RVLM neurons receive inputs from glycinergic neurons located in the ventral brainstem and a subset of the PRV-labelled neurons express GlyT2. These co-localization studies were conducted 120 h after inoculation of the kidney and we suggest that the double-labelled neurons are interneurons; however, at this time point the PRV labelled neurons contain pre-sympathetic neurons and interneurons projecting to pre-sympathetic neurons. Nevertheless, PRV is the only tool to reliably identify pre-sympathetic neurons and circuits in supraspinal areas.

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

## Additional information

### Data availability statement

All data are available upon request with specific statistical tests.

### Competing interests

The authors declare that the submitted work was carried out without the presence of any personal, professional or financial relationships that could potentially be construed as a conflict of interest.

### Author contributions

A.V.D. and A.Z. designed all experiments. A.V.D., A.Z., H.G. and L.D.D. wrote the manuscript. H.G. performed electrophysiological experiments. L.D.D., A.J.R.M. and C.M.D. performed immunofluorescence experiments, viral tracing, and imaging. H.G., L.D.D., A.J.R.M., C.M.D., A.Z. and A.V.D. edited and approved the manuscript. H.G., L.D.D., A.J.R.M., C.M.D., A.Z. and A.V.D. agree to be accountable for all aspects of the work.

### Funding

Reserch reported in this publication was supported by an Institutional Development Award (IDeA) from National Institute of General Medical Sciences and the National Institute of Diabetes and Digestive and Kidney Diseases of the National Institutes of Health under award number DK137224 and National Institute of Diabetes and Digestive and Kidney Diseases of the National Institute of Health under award number DK122842.

### Keywords

autonomic nervous system, electrophysiology, glycine, GABA, glycine transporter 2, pseudorabies virus, RVLM, VLM/VMM

### Supporting information

Additional supporting information can be found online in the Supporting Information section at the end of the HTML view of the article. Supporting information files available:

**Peer Review History**

