## [Peer Review History · The Journal of Physiology]

THE SYMPATHOREGULATORY REGION OF THE MOUSE ROSTRAL BRAINSTEM RELIES ON BOTH GABA AND GLYCINE TO GENERATE INHIBITORY CURRENTS

Hong Gao, Lucie D Desmoulins, Adrien J.R. Molinas, Courtney M Dugas, Andrea Zsombok, and Andrei V Derbenev
DOI: 10.1113/JP288116

Corresponding author(s): Andrei Derbenev (aderben@tulane.edu)

Review Timeline:

Submission Date:	15-Nov-2024
Editorial Decision:	10-Jan-2025
Revision Received:	11-Dec-2025
Accepted:	26-Jan-2026

Senior Editor: Katalin Toth

Reviewing Editor: Samuel Young

Transaction Report:

Dear Dr Derbenev,

Re: JP-RP-2024-288116 "CO-RELEASE OF GABA AND GLYCINE IN THE SYMPATHOREGULATORY REGION OF THE MOUSE ROSTRAL BRAINSTEM" by Hong Gao, Lucie D Desmoulins, Adrien J.R. Molinas, Courtney M Dugas, Andrea Zsombok, and Andrei V Derbenev

Thank you for submitting your manuscript to The Journal of Physiology. It has been assessed by a Reviewing Editor and by 2 expert referees and we are pleased to tell you that it is potentially acceptable for publication following satisfactory major revision.

REVISION CHECKLIST:

We look forward to receiving your revised submission.

Yours sincerely,

Katalin Toth
Senior Editor
The Journal of Physiology

REQUIRED ITEMS

- Author photo and profile. First or joint first authors are asked to provide a short biography (no more than 100 words for one author or 150 words in total for joint first authors) and a portrait photograph. These should be uploaded and clearly labelled together in a Word document with the revised version of the manuscript. See Information for Authors for further details.
- Your manuscript must include a complete Additional Information section, including competing interests; funding; author contributions and acknowledgements.
- The Journal of Physiology funds authors of provisionally accepted papers to use the premium BioRender site to create high resolution schematic figures. Follow this link and enter your details and the manuscript number to create and download figures. Upload these as the figure files for your revised submission. If you choose not to take up this offer, we require figures to be of similar quality and resolution. If you are opting out of this service to authors, state this in the Comments section on the Detailed Information page of the submission form. The link provided should only be used for the purposes of this submission. Authors will be charged for figures created on this premium BioRender account if they are not related to this manuscript submission.
- Please upload separate high-quality figure files via the submission form.
- Please ensure that the Article File you upload is a Word file.
- Please include an Abstract Figure file, as well as the Figure Legend text within the main article file. The Abstract Figure is a piece of artwork designed to give readers an immediate understanding of the research and should summarise the main conclusions. If possible, the image should be easily 'readable' from left to right or top to bottom. It should show the physiological relevance of the manuscript so readers can assess the importance and content of its findings. Abstract Figures should not merely recapitulate other figures in the manuscript. Please try to keep the diagram as simple as possible and

without superfluous information that may distract from the main conclusion(s). Abstract Figures must be provided by authors no later than the revised manuscript stage and should be uploaded as a separate file during online submission labelled as File Type 'Abstract Figure'. Please also ensure that you include the figure legend in the main article file. All Abstract Figures should be created using BioRender. Authors should use The Journal's premium BioRender account to export high-resolution images. Details on how to use and access the premium account are included as part of this email.

Reviewing Editor:

Comments to ensure the paper complies with the Statistics Policy:

Data needs to be presented as SD. Currently, data is presented as SEM.

Comments to the authors:

Both reviewers found the findings to be highly impactful with the potential to provide new insight into how GABA and glycine neurotransmitter release regulates the autonomic nervous system. The data is highly rigorous and the combination of a multi-disciplinary techniques is outstanding. However, both reviewers had concerns. A major concern is that while the authors definitely show presynaptic GABA and glycine release does occur, there is not conclusive evidence that this is co-release from the same terminal. To address the concern, the authors have two options, they will either need to perform additional experiments to definitively demonstrate co-release or the authors will need to rewrite their manuscript to temper these conclusions and change the manuscript title to reflect the tempered conclusions. In addition, the authors need to address the comments about the logic, timing and sequence of application of pharmacological agents. The authors need to carefully revise and rewrite their manuscript in response to the multiple positive and careful comments by the reviewers. Finally the authors need to report all data as SD and not SEM to comply with Journal policy.

Referee #1:

The current submission from Gao and colleagues examines the inhibitory regulation of presympathetic ventral medulla neurons by GABA and glycine. The manuscript identifies the release of both GABA and glycine from GlyT2-expressing cells. Further, this regulation relies on the glycine transporter. The authors also identify GlyT2-expressing cells neighboring the presympathetic ventral medulla neurons. Overall, the submission makes a substantial contribution to the existing literature on inhibitory regulation of the sympathetic nervous system by demonstrating the necessity of GABA and glycine for the regulation of presympathetic neurons. The combination of genetics, pharmacology, electrophysiology, and anatomy are a

strength. Some minor to moderate concerns relate to interpretation and the relationship between interpretation and research design.

Moderate:

The authors convincingly show that GlyT2-expressing cells functionally signal through both the GABA_A receptor and glycine receptor in neurons with polysynaptic inputs to the kidney. While they rely on prior reports that identified coexpression of GAD/vGAT with GlyT2, the current data do not directly demonstrate that the 2 transmitters are released from the same presynaptic terminal, much less the same synaptic vesicle. Although it is an important finding that GlyT2-expressing cells rely on both GABA_A and glycine receptors to generate IPSCs, the widespread conclusion throughout the paper that the authors have identified co-release is not fully supported. In the absence of electron microscopy and/or single-synapse photo-illumination, this interpretation is better suited as a possibility for discussion than the primary finding of the paper.

There are several other instances where a finding is reported but the design does not allow for alternative findings. For instance, the identification of $\alpha 1$ and $\alpha 3$ as the prevalent forms of the glycine receptor when no other receptor identities were queried. Similarly, the primary glycinergic inputs to the presympathetic ventral medulla were identified in the neighboring areas of the ventral medulla without consideration of inputs from the dorsal medulla, midbrain, or pons.

It is sometimes difficult to follow the logic of the pharmacological sequences and whether alternative conclusions were considered. For instance, strychnine is never applied as a single agent to selectively query glycinergic signaling. It is only applied after bicuculline, ORG, 4-AP, etc. Whether different conclusions could be drawn by a different sequence or the specific degree of glycine receptor dependence for eIPSCs are not considered. Furthermore, the summary figure seems to indicate more glycinergic signaling than GABAergic signaling after 4-AP but this was not investigated in the absence of bicuculline.

Minor:

The authors report the number of cells recorded and that both sexes were used but no other information is provided for the number of subjects used in any of the studies or the relative distribution of sex across outcomes.

Referee #2:

This manuscript used a combination of neuroanatomical and electrophysiological approaches to test whether presympathetic neurons in the VLM/VMM receive glycinergic inputs and if glycine release is increased when inhibitory activity increases. They found that the sIPSCs recorded from presympathetic neurons in the ventral brainstem are generated by presynaptic release of GABA and glycine and increased network activity led to increased glycine release. The anatomical study identified the location of glycinergic neurons in the ventral brainstem and they found that GlyRs expressed in presympathetic VLM/VMM neurons are formed by $\alpha 1$ and $\alpha 3$ subunits. Here are several concerns to invite the authors to consider.

1). I am not sure if authors can call this co-release... maybe released from the same neuron but not necessarily from the same terminal, which would need mini-IPSC recordings for this, block by bicuculline and strychnine, or 2photon, single spine stimulation. In addition, the consideration of the paragraph starting from Line 617 should not be tacked on the end of the discussion but appropriately addressed throughout to assuage the impression that this is co-release from the same terminal.

2). Lines 340-347: What is the timing of the application of ORG? I presume that this would need to be a prolonged application to deplete the vesicular storage of Gly. It would be nice to show that in the presence of ORG GlyR stimulation

was depleted over time, but GABAR remains intact. Why not have a strychnine IPI example? What does strychnine alone do to these responses?

3). Pharmacology of the IPI experiments should be done at more than just the one time point to show transmitter recycling/receptor desensitization better. Or is this a ChR2 limitation? Patching the pre-synaptic cell and measuring AP fidelity would be a useful control.

4). Line 590: Is the dual exponential decay discussed here? Can the data be analyzed for dual component decays, and show that it drops to a single component with pharmacological blockers?

5). Did the author correct the familywise error rate since multiple different measurements were performed from the same current response when they did student t-tests, e.g. the frequency, amplitudes, and Iphasic of IPSCs? In addition, besides p-values, it will make the MS more readable if statistical information (e.g., the statistical test used) is also included in the figure legends.

6). Although it is a very minor issue, it will be great if the digits of significant figures can be presented consistently. For example, Line-296: **21.34** {plus minus} 1.7 pA vs. **18.22** {plus minus} 1.39 pA; Line-311: **17.4** {plus minus} 2.1 pA vs. **25.9** {plus minus} 4.4 pA. It is unnecessary to show four significant figures for numbers/values less than 100, and three significant figures for numbers/values less than 10.

7). Line 323: ChR2^{EYFP} seems like a typo

END OF COMMENTS

RESPONSE TO THE REVIEWERS' COMMENTS:

We would like to thank the Reviewers for their thoughtful feedback and constructive comments on the manuscript. We have now addressed each concern raised either via clarification, additional experimentation, and/or edits to the text. We believe that the revisions made in accordance with the reviewers' recommendations have significantly improved the manuscript. Details for our specific responses are provided below with changed lines referencing the final clean version of the manuscript. We hope the manuscript is now acceptable for publication.

Reviewing Editor:

Comments to ensure the paper complies with the Statistics Policy:

Data needs to be presented as SD. Currently, data is presented as SEM.

Response: We apologize for the oversight. Now all the data are displayed as means +/- SD.

Comments to the authors:

Both reviewers found the findings to be highly impactful with the potential to provide new

insight into how GABA and glycine neurotransmitter release regulates the autonomic nervous system. The data is highly rigorous and the combination of a multi-disciplinary techniques is outstanding. However, both reviewers had concerns. A major concern is that while the authors definitely show presynaptic GABA and glycine release does occur, there is not conclusive evidence that this is co-release from the same terminal. To address the concern, the authors have two options, they will either need to perform additional experiments to definitively demonstrate co-release or the authors will need to rewrite their manuscript to temper these conclusions and change the manuscript title to reflect the tempered conclusions.

In addition, the authors need to address the comments about the logic, timing and sequence of application of pharmacological agents. The authors need to carefully revise and rewrite their manuscript in response to the multiple positive and careful comments by the reviewers. Finally, the authors need to report all data as SD and not SEM to comply with Journal policy.

Response: Thank you very much for the positive evaluation and constructive criticism. Following the reviewers' requests we have now rewritten the manuscript to temper down the conclusions and changed the manuscript title. In addition, in the method section we addressed and justified the logic and sequence of application of pharmacological agents. We also responded to all review's questions point by point.

Referee #1:

The current submission from Gao and colleagues examines the inhibitory regulation of presympathetic ventral medulla neurons by GABA and glycine. The manuscript identifies the release of both GABA and glycine from GlyT2-expressing cells. Further, this regulation relies on the glycine transporter. The authors also identify GlyT2-expressing cells neighboring the presympathetic ventral medulla neurons. Overall, the submission makes a substantial contribution to the existing literature on inhibitory regulation of the sympathetic nervous system by demonstrating the necessity of GABA and glycine for the regulation of presympathetic neurons. The combination of genetics, pharmacology, electrophysiology, and anatomy are a strength. Some minor to moderate concerns relate to interpretation and the relationship between interpretation and research design.

Moderate

Comments: The authors convincingly show that GlyT2-expressing cells functionally signal through both the GABAA receptor and glycine receptor in neurons with polysynaptic inputs to the kidney. While they rely on prior reports that identified coexpression of GAD/vGAT with GlyT2, the current data do not directly demonstrate that the 2 transmitters are released from the same presynaptic terminal, much less the same synaptic vesicle. Although it is an important finding that GlyT2-expressing cells rely on both GABAA and glycine receptors to generate IPSCs, the widespread conclusion throughout the paper that the authors have identified co-release is not fully supported. In the absence of electron microscopy and/or single-synapse photoillumination, this interpretation is better suited as a possibility for discussion than the primary finding of the paper.

Response: We thank the reviewer for the constructive comments. We agree that only electron microscopy and/or single synapse photoillumination can provide direct evidence for GABA and glycine co-localization and co-release. Therefore, we have rewritten the manuscript to temper down the interpretation of our results.

Comment: There are several other instances where a finding is reported but the design does not allow for alternative findings. For instance, the identification of $\alpha 1$ and $\alpha 3$ as the prevalent forms of the glycine receptor when no other receptor identities were queried.

Response: We apologize for the lack of clarity regarding the subunits.

GlyRs consist of a pentameric assembly of subunits that can be homo-pentameric α or heteromeric $\alpha\beta$ receptors. The heteromeric GlyRs have a $4\alpha:1\beta$ configuration (recently reviewed in Fraser 2025). Five subunits have been identified in humans and rodents from $\alpha 1$ - $\alpha 4$ and β subunits. Previous studies showed that in mice, expressions of *Glr1* and *Glrb* transcripts were predominant, whereas *Glr2*, *Glr3* and *Glr4* were expressed at lower levels (Rajalu 2009). *Glr1* was shown to be expressed in the rodent spinal cord, brainstem, cerebellum and in sub-regions of the thalamus and hypothalamus (Malosio 1991, Ceder 2024). *Glr2* is the dominant subunit during development, and its expression levels decline after birth (reviewed in Ceder 2024, Schaefer 2018b; Darwish 2023, Malosio 1991). *GlyR* $\alpha 3$ was shown to be important for central inflammatory pain sensitization (Werynska et al., 2021), somatosensory

processing (Weman 2024), and rhythmic breathing (Manzke 2010); whereas *GlyR* $\alpha 4$ is vital for correct embryonic development (Darwish 2023, Nishizono 2020) and was not detected in the adult mouse medulla (Ceder et al., 2024). Expression of *Glr*b is widespread in the nervous system and is an important part of the heteromeric GlyRs and aids in synapse stabilization. Based on these previous findings, we originally aimed to identify GlyR $\alpha 1$ (abundant expression) and $\alpha 3$ (low expression but in the ventral brainstem). In the revised manuscript we have provided evidence that β subunits are also expressed in the ventral brainstem (new Figure 7). Previous data show that $\alpha 2$ and $\alpha 4$ are primarily important for development; therefore, in this study we did not determine their expression levels. Methods and results sections were modified accordingly (Page 11, 18-19).

Comment: Primarily, the primary glycinergic inputs to the presympathetic ventral medulla were identified in the neighboring areas of the ventral medulla without consideration of inputs from the dorsal medulla, midbrain, or pons.

Response: We apologize for not being clear. The location of glycinergic inputs to presympathetic VLM/VMM neurons were determined using retrograde AAV tracing from the VLM. We found GFP positive fibers in many brain areas including the lateral preoptic area, lateral hypothalamus (Response Fig 1A), the thalamus, the red nucleus (Response Fig 1B), the superior colliculus, the lateral lemniscus, the inferior olive and the reticular nucleus from the anterior formation to the medullary part; however, GFP positive neurons were only found in the area surrounding the VLM/VMM. Our findings suggest that presympathetic VLM/VMM neurons receive glycinergic inputs mainly from

Response Figure 1: Representative images showing GFP positive fibers and neurons. GFP positive fibers in the lateral hypothalamus (A), red nucleus and superior colliculus (B) and cells in the intermediate reticular nucleus.

the surrounding area; therefore, other areas including the dorsal medulla, midbrain, and pons, were not included. The results section (page 18) and discussion (page 23) were revised accordingly.

Comment: It is sometimes difficult to follow the logic of the pharmacological sequences and whether alternative conclusions were considered. For instance, strychnine is never applied as a single agent to selectively query glycinergic signaling. It is only applied

after bicuculline, ORG, 4-AP, etc. Whether different conclusions could be drawn by a different sequence or the specific degree of glycine receptor dependence for eIPSCs are not considered.

Response: We apologize for not being clear. Pharmacological sequences of our experiments were designed based on evidence that strychnine blocks GABA_ARs (Braestrup and Nielsen, 1980). Therefore, to eliminate interaction of strychnine with GABA_ARs, first we blocked GABA_ARs with bicuculline, then applied strychnine to confirm GlyRs involvement. In the revised version of the manuscript, we have included clarification about the pharmacological sequences of the experiments (materials and methods, page 9).

Furthermore, based on the reviewer's comment we have conducted additional experiments to determine the effect of strychnine on sIPSC frequency in the concentration which was used in our experiments (Response Fig2). Our preliminary findings suggest that strychnine can inhibit synaptic currents in pre-sympathetic neurons to a higher degree than previously anticipated. Application of strychnine (1 μ M) significantly reduced the average frequency of sIPSCs from 1.8 ± 2.52 Hz to 0.2 ± 0.18 Hz ($n = 6$, Wilcoxon matched-pairs signed-rank test, $P = 0.032$). This is a potentially important finding requiring future investigations including dose-response curve to determine the effect of strychnine on this particular group of neurons. Please note that this finding does not disagree with our overall findings that glycinergic inputs simultaneously release GABA and glycine and inhibition of postsynaptic neurons rely on both GABA_ARs and GlyRs.

Response Figure 2. Application of strychnine (1 μ M) resulted in inhibition of sIPSCs in pre-sympathetic kidney-related RVLN neurons, suggesting that strychnine can inhibit both GABAergic and glycinergic inhibitory currents.

Comment: Furthermore, the summary figure seems to indicate more glycinergic signaling than GABAergic signaling after 4-AP but this was not investigated in the absence of bicuculline.

Response: Thank you for this important point. The summary figure (graphical abstract) was updated.

In our previous study in rats (Gao et. al., 2019) we demonstrated that applications of 4-AP increased the overall frequency of spontaneous IPSCs and subsequent application of bicuculline decreased the frequency of sIPSCs by eliminating GABAergic neurotransmission. We found significant difference between sIPSC

frequency in the presence of bicuculline and in the presence of 4-AP plus bicuculline. We also showed that increased activity of synaptic inputs (by application of 4-AP) decreased the percentage of GABAergic and increased the percentage of glycinergic sIPSCs. Therefore, we concluded that GABA is a major inhibitory neurotransmitter under steady-state conditions, whereas release of glycine requires the potentiation of active synaptic inputs. In the current study, we aimed to demonstrate that increased activity of inhibitory synaptic inputs to presympathetic neurons potentiates release of glycine; therefore, we separated glycinergic and GABAergic spontaneous IPSCs with application of bicuculline. We found that in the presence of bicuculline, application of 4-AP increased the frequency of sIPSCs suggesting increased glycine release. These data suggest that release of glycine requires higher degree of activation of presynaptic neurons.

Minor

Comment: The authors report the number of cells recorded and that both sexes were used but no other information is provided for the number of subjects used in any of the studies or the relative distribution of sex across outcomes.

Response: We agree with the reviewer and compared the frequency of sIPSCs (Fig. 1) in males and females. The mean sIPSC frequency of presympathetic VLM/VMM neurons in male mice was 3.501 ± 2.489 Hz (n=10), whereas 1.859 ± 2.621 Hz (n=15) in female mice. The analysis showed an increasing trend in males, but without significance (p=0.0709, Mann Whitney test). Similarly, there was no significant difference in sIPSC amplitude between males and females (20.84 ± 8.364 pA vs. 21.79 ± 8.305 pA, p=0.7836, unpaired t-test). After bath application of bicuculline there was no difference in the frequency and amplitude of sIPSCs (1.876 ± 2.427 Hz vs. 1.302 ± 1.968 Hz, p=0.1438, Mann Whitney test; 19.55 ± 7.591 pA vs. 17.27 ± 6.328 pA, p=0.4327, unpaired t-test; n=10 male and n=15 female).

We also compared the amplitude of eIPSCs (Fig 2.) in male and female mice. The amplitude of eIPSCs was 474.6 ± 354.1 pA in males (n=13) and 592.1 ± 589.0 pA in females (n=11), p=0.9095, Mann-Whitney test.

In summary, our data did not reveal significant sex-dependent difference in the frequency and amplitude of sIPSCs and eIPSCs; therefore, the data from male and female mice were pooled. We also included this information into the discussion (page 21).

Referee #2:

This manuscript used a combination of neuroanatomical and electrophysiological approaches to test whether presympathetic neurons in the VLM/VMM receive glycinergic inputs and if glycine release is increased when inhibitory activity increases. They found that the sIPSCs recorded from presympathetic neurons in the ventral brainstem are generated by presynaptic release of GABA and glycine and increased network activity led to increased glycine release. The anatomical study identified the location of glycinergic neurons in the ventral brainstem, and they found that GlyRs expressed in presympathetic VLM/VMM neurons are formed by $\alpha 1$ and $\alpha 3$ subunits. Here are several concerns to invite the authors to consider.

Comment: I am not sure if authors can call this co-release... maybe released from the same neuron but not necessarily from the same terminal, which would need mini-IPSC recordings for this block by bicuculline and strychnine, or 2photon, single spine stimulation.

Response: We agree with the reviewer that we cannot call it co-release; therefore, we have made appropriate changes to the manuscript and tempered down the interpretation of the results.

Comment: In addition, the consideration of the paragraph starting from Line 617 should not be tacked on the end of the discussion but appropriately addressed throughout to assuage the impression that this is corelease from the same terminal.

Response: We agree with the reviewer and the manuscript has been rewritten to better reflect the findings.

Comment: Lines 340-347: What is the timing of the application of ORG? I presume that this would need to be a prolonged application to deplete the vesicular storage of Gly. It would be nice to show that in the presence of ORG GlyR stimulation was depleted over time, but GABAR remains intact.

Response: Thank you for pointing this out. In our experiments brain slices were incubated for 10 minutes in ACSF containing 10 μM of ORG. To determine the effect of ORG on evoked IPSCs, we conducted additional experiments. Our data show that ORG alone significantly decreased the amplitude of light evoked IPSCs (Response Figure 3) in 8 out of 12 neurons. This suggests that blockade of GlyT2 with ORG depletes glycine.

On the other hand, in one cell ORG application did not affect the amplitude of eIPSCs, whereas in three recorded neurons increased eIPSC amplitude. This is somewhat expected, because it is unlikely that the vesicles will stay empty after ORG application, and GABA and glycine may substitute for each other if vesicular concentration of the neurotransmitters changes (Aubrey et al., 2007; Edwards, 2007). This is supported by data from rat trigeminal nucleus showing that 60% of vesicles contained GABA and glycine, 17% contained only GABA and glycine alone was found in 18% of the vesicles.

Comment: Why not have a strychnine IPI example? What does strychnine alone do to these responses?

Response: The design of our experiments was based on evidence that strychnine blocks GABA_ARs (Braestrup and Nielsen, 1980). Therefore, to eliminate interaction of strychnine with GABA_ARs, first we blocked GABA_ARs with bicuculline, then applied strychnine to confirm GlyRs involvement. In the revised version of the manuscript, we have included clarification about the pharmacological sequences of the experiments (materials and methods, page 9). Please, also see Response Figure 2 above.

Comment: Pharmacology of the IPI experiments should be done at more than just the one time point to show transmitter recycling/receptor desensitization better. Or is this a ChR2 limitation? Patching the pre-synaptic cell and measuring AP fidelity would be a useful control.

Response: We used this single time point because based on our experience with recordings from RVLN neurons the time interval of 50 msec between the light pulses is the best time to study efficiency/reliability of neurotransmitter release. After the first response (peak 1 and 2), there is no significant change in amplitude (peaks 2-4). This suggests that the number of neurotransmitters in the vesicles is decreasing, but the system maintains a steady-state level.

We agree with the reviewer that patch-clamp recordings from presynaptic neurons (inhibitory) and measuring the fidelity of action potentials would be useful control. To conduct these types of experiments in the VLM/VMM is extremely challenging. It would require identification of presynaptic glycinergic input neurons and additional identification of kidney-related pre-sympathetic neurons. Although the findings would be exciting, conducting these studies would require additional time and resources.

Comment: Line 590: Is the dual exponential decay discussed here? Can the data be analyzed for dual component decays, and show that it drops to a single component with pharmacological blockers?

Response: We have analyzed the data for dual component decays. Overall, the weighted decay time for sIPSCs was 17.24 ± 19.79 ms, whereas after bicuculline application it was 12.46 ± 10.33 ms ($n=24$; $p=0.2405$). In the control condition, 11 cells showed a single-exponential decay, and 13 cells showed a double-exponential decay. After bath application of bicuculline, 5 out of the 11 cells showing double-exponential decay switched to a single-exponential decay. This information was included in the results (Page 13).

Comment: Did the author correct the familywise error rate since multiple different measurements were performed from the same current response when they did student t-tests, e.g. the frequency, amplitudes, and Iphasic of IPSCs? In addition, besides p-values, it will make the MS more readable if statistical information (e.g., the statistical test used) is also included in the figure legends.

Response: Thank you for pointing this out. To control family-wise error we employed Holm-Bonferroni method. This sequential method is a less conservative, but more powerful because it rejects more hypotheses while still maintaining the desired overall error rate. This information was included in the results Holm-Bonferroni adjusted P value was included in the results.

Comment: Although it is a very minor issue, it will be great if the digits of significant figures can be presented consistently. For example, Line-296: **21.34** {plus minus} 1.7 pA vs. **18.22** {plus minus} 1.39 pA; Line-311: **17.4** {plus minus} 2.1 pA vs. **25.9** {plus minus} 4.4 pA. It is unnecessary to show four significant figures for numbers/values less than 100, and three significant figures for numbers/values less than 10.

Response: Thank you for pointing this out, we have now updated the data.

Comment: Line 323: ChR2EYFP seems like a typo

Response: EYFP (enhanced yellow fluorescent protein) is a genetically engineered variant of EGFP that has been mutated to shift its excitation and emission to longer wavelengths. Upon excitation the EYFP emits yellow light. The peak emission for EYFP which is 504 nm is very close to peak emission of EGFP (enhanced green fluorescent protein) which is 527 nm. In our study we used EGFP/FITC/Cy2/AlexaFluor 488 bandpass filter to identify expression of ChR2^{EYFP}. The yellow fluorescence appeared as green.

Dear Dr Derbenev,

Re: JP-RP-2025-288116R1 **"THE SYMPATHOREGULATORY REGION OF THE MOUSE ROSTRAL BRAINSTEM RELIES ON BOTH GABA AND GLYCINE TO GENERATE INHIBITORY CURRENTS"** by Hong Gao, Lucie D Desmoulins, Adrien J.R. Molinas, Courtney M Dugas, Andrea Zsombok, and Andrei V Derbenev

We are pleased to tell you that your paper has been accepted for publication in The Journal of Physiology.

Yours sincerely,

Katalin Toth
Senior Editor
The Journal of Physiology

IMPORTANT POINTS TO NOTE FOLLOWING ACCEPTANCE OF YOUR PAPER:

- **IMPORTANT NOTICE ABOUT OPEN ACCESS:** To assist authors whose funding agencies mandate immediate public access to published research findings, The Journal of Physiology allows authors to pay an Open Access (OA) fee to have their papers made freely available immediately on publication.

- You can help your research get the attention it deserves! Check out Wiley's free Promotion Guide for best-practice recommendations for promoting your work at: www.wileyauthors.com/eeo/guide. You can learn more about Wiley Editing Services which offers professional video, design, and writing services to create shareable video abstracts, infographics, conference posters, lay summaries, and research news stories for your research at: www.wileyauthors.com/eeo/promotion.

- If you would like to receive our 'Research Roundup', a monthly newsletter highlighting the cutting-edge research published in The Physiological Society's family of journals (The Journal of Physiology, Experimental Physiology, Physiological Reports, The Journal of Nutritional Physiology and The Journal of Precision Medicine: Health and Disease), please click this link, fill in your name and email address and select 'Research Roundup': <https://www.physoc.org/journals-and-media/membernews>

EDITOR COMMENTS

Reviewing Editor:

The authors have done a good job of responding to the previous critiques. There are no further concerns.

REFEREE COMMENTS

Referee #1:

The authors have appropriately tempered conclusions so that they are supported by the data. Additional concerns were thoroughly justified by the authors providing additional information and references.

Referee #2:

Thank you for carefully considering my comments for the revision. I have one very minor comment on the the digits of significant figures I brought it up in the previous review. The digits of significant figures can be presented consistently.